# Small molecule regulators of microRNAs identified by high-throughput screen coupled with high-throughput sequencing

Lien D. Nguyen [1,6], Zhiyun Wei [1,2,6] ✉, M. Catarina Silva [3], Sergio Barberán-Soler[4], Jiarui Zhang[5], Rosalia Rabinovsky[1], Christina R. Muratore[1], Jonathan M. S. Stricker[1], Colin Hortman[4], Tracy L. Young-Pearse [1], Stephen J. Haggarty [3] & Anna M. Krichevsky [1] ✉

MicroRNAs (miRNAs) regulate fundamental biological processes by silencing mRNA targets and are dysregulated in many diseases. Therefore, miRNA replacement or inhibition can be harnessed as potential therapeutics. However, existing strategies for miRNA modulation using oligonucleotides and gene therapies are challenging, especially for neurological diseases, and none have yet gained clinical approval. We explore a different approach by screening a biodiverse library of small molecule compounds for their ability to modulate hundreds of miRNAs in human induced pluripotent stem cell-derived neurons. We demonstrate the utility of the screen by identifying cardiac glycosides as potent inducers of miR-132, a key neuroprotective miRNA downregulated in Alzheimer's disease and other tauopathies. Coordinately, cardiac glycosides downregulate known miR-132 targets, including Tau, and protect rodent and human neurons against various toxic insults. More generally, our dataset of 1370 drug-like compounds and their effects on the miRNome provides a valuable resource for further miRNA-based drug discovery.

Messenger RNAs (mRNA) have recently emerged as promising targets for numerous disease categories, with several approved mRNA therapeutics in the last five years[1]. However, >70% of the human genome is transcribed into noncoding RNAs (ncRNAs), many of which play essential yet largely understudied roles in biological processes[2,3]. Among ncRNAs, microRNAs (miRNAs) are established critical regulators of gene expression that facilitate the degradation and inhibit the translation of mRNA targets[4]. Specific miRNAs have been shown to be dysregulated in various diseases[5], making them valuable targets for both diagnostic and therapeutic purposes. Nevertheless, no miRNA-

modulatory compounds have been approved for any clinical indication.

Two common approaches to modulating miRNAs, oligonucleotide-based and gene therapy, have serious limitations. Oligonucleotide miRNA mimics and inhibitors must be heavily modified to avoid rapid degradation, often have poor intracellular delivery and off-target activity, and can induce immunotoxicity[6–8]. Similarly, delivering genes coding for miRNAs or "sponging" miRNAs through viral or non-viral vectors is generally inefficient and can induce immunotoxicity or off-target integration[7]. The central nervous system

[1]Department of Neurology, Brigham and Women's Hospital and Harvard Medical School, Boston, MA02115, USA. [2]Shanghai Key Laboratory of Maternal Fetal Medicine, Shanghai Institute of Maternal-Fetal Medicine and Gynecologic Oncology, Shanghai First Maternity and Infant Hospital, School of Medicine, Tongji University, Shanghai 200092, China. [3]Chemical Neurobiology Laboratory, Center for Genomic Medicine, Department of Neurology, Massachusetts General Hospital and Harvard Medical School, Boston, MA02114, USA. [4]RealSeq Biosciences, Santa Cruz, CA 95060, USA. [5]Division of Computational Biomedicine, Boston University School of Medicine, Boston, MA02118, USA. [6]These authors contributed equally: Lien D. Nguyen, Zhiyun Wei. ✉e-mail: zhiyun_wei@163.com; akrichevsky@bwh.harvard.edu

(CNS) presents additional challenges for drug delivery and efficacy due to the blood-brain barrier (BBB) that blocks the entrance of most compounds. We proposed small molecules as an alternative approach for modulating miRNAs[9]. Compared to miRNA oligonucleotides and gene therapy, small molecules can be optimized for better brain and cell penetrance. Small molecules already approved for treating human diseases have well-established safety profiles and pharmacokinetics. Repurposing or improving these compounds for modulating endogenous miRNA expression would accelerate the development of miRNA therapeutics. However, few systematic efforts have been made to identify such miRNA modulators, and only a small number of miRNA-modulating small molecules have been described[10–15]. Specifically, no miRNome-wide high-throughput screen (HTS) for small molecule modulators of miRNA expression or activity has been developed to date.

We designed a pipeline for discovering small molecules that regulate miRNAs in human induced pluripotent stem cell (iPSC)-derived excitatory neurons. Whereas previous screens focused on a specific miRNA[10,11,13,15] or favored a particular mechanism of action such as direct binding[12,15], our phenotypic screen is relevant to all miRNAs and inclusive to all mechanisms of action, including direct binding, transcriptional and post-transcriptional modulations, and indirect regulations. Furthermore, instead of utilizing reporter assays[10,11,13], we employed miRNA-sequencing that enabled direct expression profiling of 338 miRNAs for 1370 small molecule compounds. The dataset provides a resource for identifying candidate compounds that regulate a specific miRNA, or miRNAs regulated by a class of compounds.

To validate the screen results, we focused on miR-132, one of the most consistently downregulated miRNAs in the cortical and hippocampal neurons of patients with Alzheimer's Disease and Related Dementias (ADRD)[16–20]. miR-132 deficiency promotes Aβ plaque deposits[21,22] and Tau accumulation, phosphorylation, and aggregation[22–25]. Correspondingly, miR-132 mimics or miR-132 viral overexpression provided neuroprotection in several cellular and animal ADRD models[20,21,25,26], supporting miR-132 upregulation as a therapeutic strategy for ADRD and other tauopathies. Here we demonstrated that cardiac glycosides, which are sodium-potassium (Na+/K+) ATPase pump inhibitors, upregulated miR-132 in the nM range. Treating rodent and human neurons with nM cardiac glycosides protected neurons against various toxic insults and downregulated Tau and other miR-132 targets. Overall, we identified small molecule compounds that upregulated the neuroprotective miR-132 in neurons and provided a pipeline for discovering small molecule compounds that regulate other miRNAs for therapeutic purposes.

## Results

### Optimization of the high-throughput screen on human NGN2-iNs

We used human neurogenin 2 (NGN2)-driven iPSC-derived neurons (NGN2-iNs) as a physiologically relevant cell-based screening platform to investigate neuronal miRNome and focused on an essential neuron-enriched and neuroprotective miR-132. iPSC lines generated from donors were utilized for direct differentiation through NGN2 over-expression into excitatory neurons based on established protocols (Fig. 1a)[27]. These cells closely mimic the transcriptome and function of human neurons ex vivo and can be scaled and reproducibly employed in multiple assays[27]. Among 36 NGN2-iN lines obtained from the Religious Orders Study/Memory and Aging Project (ROS-MAP) cohort, 25 lines from donors without cognitive impairment were considered (Supplementary Fig. S1A). The transcriptomes of these lines were previously profiled[27]. The BR43 line, from an 89 year old female donor, was selected for the screen based on its median expression of major miR-132 targets, including *GSK3β, EP300, RBFOX1, CAPN2, FOXO3, TMEM106B*, and *MAPT* (Supplementary Fig. S1B). BR43 NGN2-iNs also had the lowest variation of baseline miR-132 expression among the

replicate cultures and exhibited miR-132 upregulation by the known inducers BDNF and forskolin (Supplementary Fig. S1C)[28–30].

Several steps of NGN2-iN culture and RNA collection were optimized for HTS to maximize neuronal health, lysing efficiency, and RNA yield. The protocol was tested for compatibility with small RNA-seq using the RealSeq ultra-low input system, long RNA RT-qPCR using the PrimeScript system, and small RNA RT-qPCR using the miRCURY system (Supplementary Fig. S1D, E), supporting its application in diverse quantitative RNA-based assays.

### Screen for small molecule regulators of microRNAs

Day 4 NGN2-iNs were plated onto 25 Matrigel-coated 96-well plates and differentiated into neurons, as verified by NeuN and Tau expression (Fig. 1a). On day 19, the Selleckchem library (N = 1902 compounds), a diverse library of bioactive molecules, was pin-transferred into plates to achieve 10 μM final concentration. DMSO (0.1% final concentration, N = 42) and forskolin (10 μM, N = 25) were used as the negative and positive controls, respectively. Besides the controls, the initial screen was performed with N = 1 for each compound. NGN2-iNs were imaged to monitor neuronal health 24 h later, followed by direct lysis to release RNA. Among all wells with test compounds, 324 (17.0%) were excluded because of cell death, neurite degeneration, loss of cells during washes, or enrichment of astrocytes. RNA lysates of the remaining wells were used for RealSeq small RNA library preparation designed for ultra-low input without RNA purification[31]. RealSeq libraries from each set of four 96-well culture plates were indexed with 384 multiplex barcodes and pooled for deep sequencing. After miRNA annotation, wells with less than 1,000 total annotated read counts were excluded from further analysis (N = 169, 10.7%). On average, 55,529 miRNA reads were counted per sample, and 455, 240, 182, and 64 miRNA species per sample were detected with minimal read counts of 1, 5, 10, and 100, respectively (Fig. 1b). Numerous neuronal miRNAs, such as miR-26a, miR-7, miR-191, miR-124, and miR-9/9* were abundant in DMSO-treated control NGN2-iNs (Fig. 1c). miR-132 was consistently detected and ranked among the 30 most abundant miRNAs. We further determined the top housekeeping neuronal miRNAs by calculating the coefficient of variation (COV) for each miRNA within each batch of RNA-seq and identified the miRNAs with the smallest COVs, including miR-103a/b, miR-107, and miR-191 (Fig. 1d). Figure 1e showed the miR-132 waterfall plot for 221 compounds in a 384-well plate.

### miRNome-scale HTS dataset as a resource to study miRNA-small molecule relationships

Across 6 batches, we generated miRNA profiles for 1437 samples, comprising 42 DMSO samples, 25 forskolin samples, and 1370 small molecule compounds (Supplementary Data 1–3). Each compound was annotated with a brief description, clinical indication or clinical trial status if applicable, pathway, BBB permeability, and target or compound class. We used ComBat algorithm[32] to minimize variation among batches (Supplementary Fig. S2) before performing the analyses summarized in Fig. 2.

The dataset is valuable for investigating diverse aspects of the relationship between small molecule compounds and miRNA expression. One potential application is the exploration of small molecule compounds that modulate a particular miRNA of interest. Figure 2a shows several miRNAs that are potential therapeutic targets for neurological diseases and the top 5 hits from the HTS (Fig. 2a, Supplementary Data 4). For instance, considering miR-132's neuroprotective role and its downregulation in neurodegenerative diseases, drugs that upregulate miR-132 could offer therapeutic benefits. Ouabain and digoxin, both cardiac glycosides, were the top candidates for upregulating miR-132. Notably, a subset of top hits (Fig. 2a, in bold) was common across multiple miRNAs, implying potential shared targets or beneficial effects on multiple biological processes.

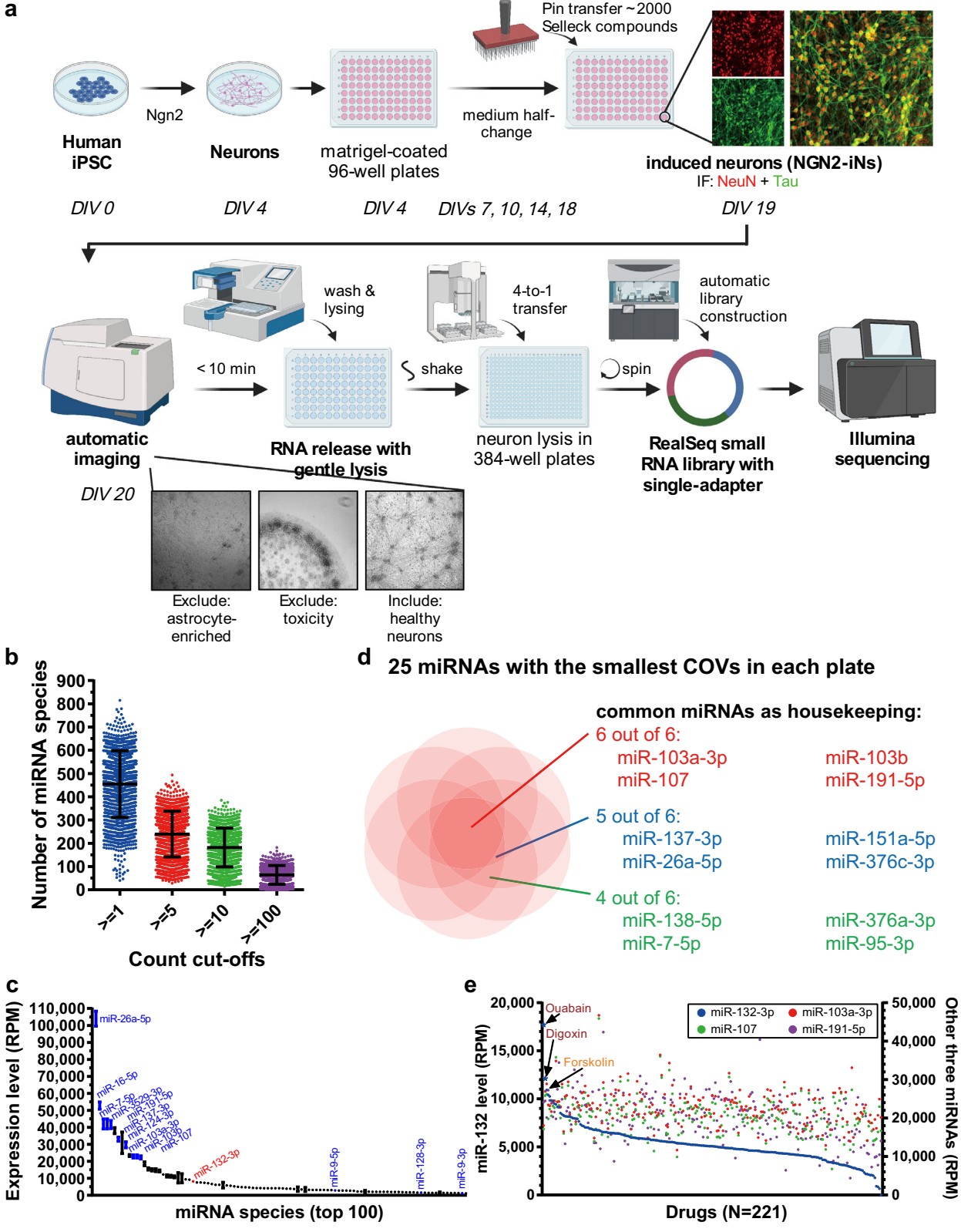

**Fig. 1 | Experimental workflow and overview of screen results in NGN2-iNs.**
**a** NGN2-iN generation, compound treatment, and miRNA-seq workflow ($N = 1$ per compound, 1902 compounds in total, created with BioRender.com). **b** Average number of miRNA species detected per sample by miRNA-seq at various count cut-offs ($N = 1371$ samples). Error bars represent mean +/− SD. **c** Expression levels of the 100 most abundant miRNAs in vehicle-treated samples. miR-26a-5p was the most abundant miRNA detected, miR-132-3p was the 27th ($N = 42$ DMSO-treated samples). Error bars represent mean +/− SEM. **d** Shared miRNAs with the lowest coefficient of variation in the plates tested. **e** Waterfall plot for miR-132 expression in plate 2 was shown in blue. Samples treated with ouabain, digoxin, and the positive control forskolin showed the highest level of miR-132. miR-107, miR-103a-3p, and miR-191-5p were shown in red, green, and purple, respectively.

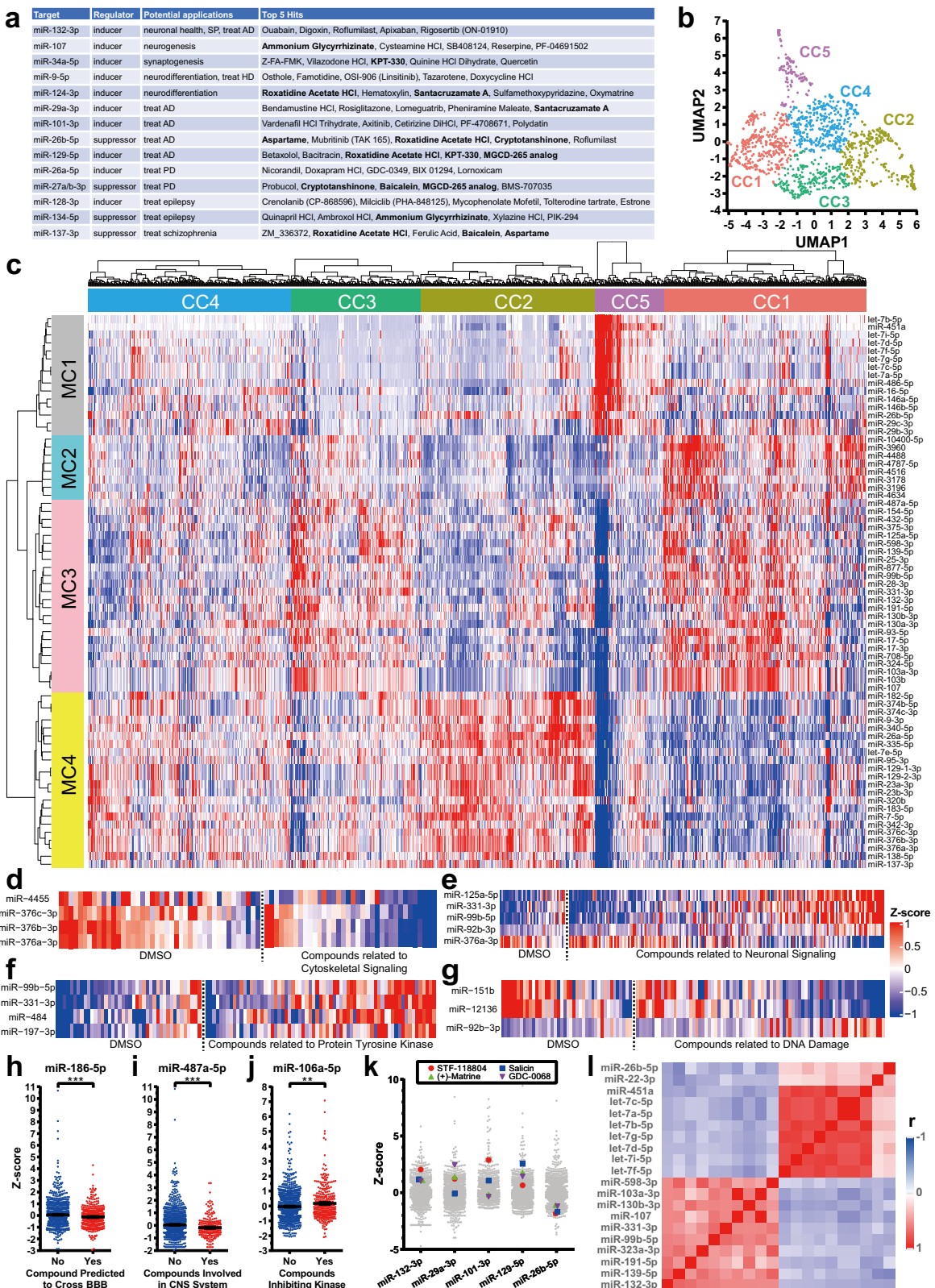

We then explored whether different compounds could elicit similar effects on the miRNome. Using UMAP algorithm for unsupervised dimension reduction, all compounds were categorized into 5 compound clusters (CCs) (Fig. 2b). The signature miRNAs most responsible for cluster organization ($N = 69$) were identified through a linear model (FDRq<0.0001) and grouped into 4 hierarchical miRNA clusters (MC1-MC4). As shown in Fig. 2c, CC5 induced MC1 and suppressed MC3. Both MC2 and MC3 were upregulated by CC1. CC5 uniquely downregulated MC3, while CC2 elevated MC4 levels. Further dissection of the CCs led to the identification of specific enrichment pathways. For example, CC5 was enriched in compounds modulating cytoskeletal signaling (>4 fold, $p = 3.9E-6$, $\chi^2$-test), some of which inhibited miR-4455 and miR-376 family ($p < 0.001$, t-test; Fig. 2d). In contrast, CC1 exhibited enrichment in compounds associated with

**Fig. 2 | Landscape of compound effects on miRNome in NGN2-iNs. a** Top small molecule hits for miRNAs with neurologic relevance. **b** UMAP analysis of miRNomes affected by compounds ($N = 1437$ samples). **c** Clustered heatmap illustrating small molecule compounds (CC1-CC5) as modulators of miRNAs (MC1-MC4). Heatmaps showing miRNAs regulated by compounds related to cytoskeletal signaling ($N = 42$ vs 36) (**d**), neuronal signaling ($N = 42$ vs 204) (**e**), protein tyrosine kinase ($N = 42$ vs 67) (**f**), and DNA damage ($N = 42$ vs 80) (**g**), respectively ($p < 0.001$, t-test). Representative miRNAs affected by compounds predicted to cross the BBB ($N = 948$ vs

447) (**h**), involved in CNS system ($N = 1145$ vs 250) (**i**), and kinase inhibitors ($N = 1062$ vs 333) (**j**). $p = 0.0004$, 9.8E−5, and 0.0021, respectively; two-sided t-test without assumption of equal SD. **k** Compounds simultaneously upregulating multiple "neuroprotective" and downregulating "neurotoxic" miRNAs. **l** miRNAs showing the strongest positive and negative correlations with miR-132 in all sequenced samples. **p < 0.01; ***p < 0.001, unpaired two-tailed Student's t test; CC compound cluster, MC miRNA cluster, AD Alzheimer's disease, HD Huntington's disease, PD Parkinson's Disease, SP synaptic plasticity, BBB blood-brain barrier.

neuronal signaling ($p = 0.003$), characterized by distinct expression of 5 miRNAs (Fig. 2e). CC3 was enriched in compounds related to protein tyrosine kinase ($p = 0.018$), generally inducing expression of miRs-99b, −197, −331 and −484 (Fig. 2f). Compounds linked to DNA damage consistently upregulated miR-92b and downregulated miRs-151b and −12136 (Fig. 2g). Compounds predicted to cross the BBB correlated with lower levels of miR-186-5p (Fig. 2h), a miRNA implicated in the blood-brain tumor barrier[33]. Compounds "involved in CNS system" downregulated miR- 487a-5p (Fig. 2i), a primate-specific miRNA largely of unknow function that was found to be dysregulated in AD[19]. Additionally, miR-106a-5p was commonly induced by kinase inhibitors (Fig. 2j), suggesting involvement of protein phosphorylation in its regulation.

Considering synergistic miRNA-mediated neuroprotection, we identified compounds that concurrently induced expression of miRs-132, −29a, −101, −129 and inhibited −26b, modulations known to protect neurons against toxic insults (Fig. 2a, Supplementary Data 4). Four compounds, STF-118804, salicin, (+)-matrine, and GDC-0068, met these criteria, albeit with modest impacts on individual miRNAs (Fig. 2k). Interestingly, salicin was reported as neuroprotective, promoting neurite[34,35], and (+)-matrine was found to enhance neural circuit remodeling and axonal growth[36,37]. Furthermore, miR-26b, a miRNA promoting neuronal apoptosis[38], was negatively correlated with neuroprotective miR-132 (Fig. 2l). The let-7 family, co-clustered with miR-26b into MC1 and found to be neurotoxic[39] or upregulated in AD[19], was also negatively correlated with miR-132 (Fig. 2l). This mutually exclusive expression pattern hints at an undiscovered mechanism coordinating distinct populations of miRNAs linked to neuroprotection and neurotoxicity.

### Screen validation: cardiac glycosides upregulate miR-132 transcriptionally and specifically

We selected miR-132, a well-established neuroprotective miRNA, to explore the utility of the miRNome-scale HTS dataset. To select candidate compounds for miR-132 upregulation, we used miR-132 plate rank as the primary criterion and adjusted with secondary criteria, including clinical approval, BBB penetrance, clinical trials, published data on neuroprotective effects, and effects on other miRNAs. As miR-132 is fully conserved, and many of its targets are highly conserved across mammals (Supplementary Fig. S3), we also utilized rat cortical neurons for validation. We treated DIV14 primary rat cortical neurons and DIV21 human NGN2-iNs with 10 μM of 44 selected compounds and monitored miR-132 expression by RT-qPCR. Besides forskolin, 12 and 10 compounds significantly upregulated miR-132 in primary rat neurons after 24 h and 72 h, respectively, and 4 compounds significantly upregulated miR-132 in NGN2-iNs after 24 h (Fig. 3a, Supplementary Data 5). The validated miR-132 inducers included 3 of the top 5 hits identified (Fig. 2a). Notably, the cardiac glycosides, ouabain and digoxin, which inhibit Na+/K+ pumps, upregulated miR-132 in all conditions. Of note, digoxin is clinically approved for treating various heart conditions, whereas ouabain is not clinically approved or utilized in recent clinical trials.

To investigate the dose response, we selected forskolin as the positive control, digoxin, ouabain, BIX02188, nitazoxanide, and pelitinib as the hits. We included 6 additional cardiac glycosides (digitoxin,

oleandrin, bufalin, bufotalin, cinobufagin, and proscillaridin A), istaroxime - a non-cardiac glycoside that also inhibits Na+/K+ pumps[40], and BIX02189 - an analog of BIX02188. These compounds represent diverse chemical groups and mechanisms of action (Fig. 3b and Supplementary Data 6). DIV14 primary rat cortical neurons were treated with doses ranging from 1 nM to 100 μM for 24 h. Remarkably, all 8 cardiac glycosides upregulated miR-132 2.5-3-fold in the nM range, with proscillaridin A having the lowest $EC_{50}$ of 3.2 nM (Supplementary Data 6). Other compounds also dose-dependently upregulated miR-132 but with higher $EC_{50}$. For all compounds tested, miR-212, which shares the seed sequence with miR-132 and is co-expressed from the same locus[41], was similarly upregulated at almost identical $EC_{50}$, suggesting that the mechanism was largely transcriptional (Supplementary Fig. S4A, B and Supplementary Data 6). The cardiac glycosides proscillaridin A, oleandrin, digoxin, ouabain, and bufalin also upregulated miR-132 and miR-212 in a dose-dependent manner in human NGN2-iNs in the nM range (Supplementary Fig. S4C, d and Supplementary Data 6). However, BIX02188, which robustly upregulated miR-132 in primary rat neurons, had no consistent effect on miR-132 in NGN2-iNs (Supplementary Fig. S4C, D), suggesting potential differences between the two cell models.

To investigate the specificity of miR-132 upregulation by oleandrin and BIX02188, we measured the expression level of 10 other abundant neuronal miRNAs in rat primary cortical neurons after 24 h treatment. When normalized to the geometric mean of all 12 miRNAs[42], only miR-132 and miR-212 were upregulated (Supplementary Fig. S5A). The primary stem-loop transcript pri-miR-132 was also upregulated by forskolin, BIX02118, and the cardiac glycosides (Fig. 3c), suggesting that these compounds activated the transcription of the miR-132/212 locus. CREB is a known transcriptional activator of the miR-132/212 locus[28–30], and we hypothesized that the identified compounds regulate miR-132/212 via CREB. Indeed, the upregulation of miR-132 by various compounds was completely blocked by pretreatment with the transcription inhibitor actinomycin D or a CREB inhibitor (Fig. 3d, e and Supplementary Fig. S5B, D). We further tested oleandrin and proscillaridin A combinations and observed no additional synergistic effects on miR-132/212 level (Supplementary Fig. S5E). In contrast, combinations of oleandrin or proscillaridin A with forskolin led to additional synergistic upregulation, suggesting that cardiac glycosides and forskolin act through non-identical, if possibly overlapping, mechanisms. As cardiac glycosides are conventional inhibitors of Na+/K+ pumps, we also knocked down ATP1A1 and ATP1A3, the dominant isoforms in neurons, with siRNAs. Knocking down either ATP1A1 or ATP1A3 also increased the expression of products of the miR-132/212 locus in rat primary neurons (Fig. 3f and Supplementary Fig. S5F), suggesting that cardiac glycosides upregulated miR-132 by inhibiting their conventional targets.

### Cardiac glycosides reduce miR-132 targets and protect against toxic insults in rodent neurons

We focused on cardiac glycosides due to their potency, efficacy, and novelty as miR-132 inducers. Furthermore, as multiple lines of evidence supported that cardiac glycoside acted through the same mechanisms, we selected oleandrin as the representative cardiac glycoside. To investigate the kinetics of miR-132/212 upregulation, we treated primary rat cortical neurons with 100 nM oleandrin and measured the

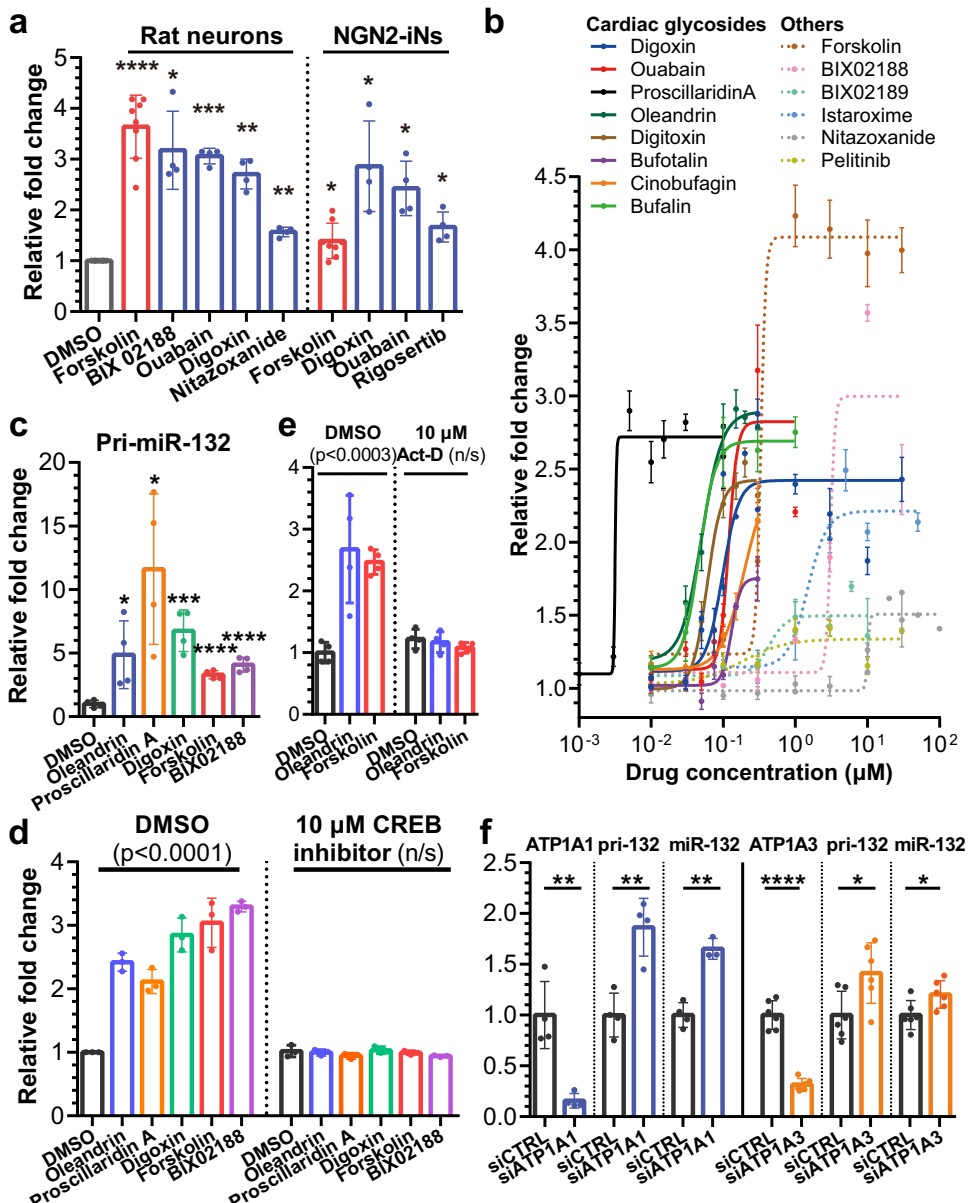

**Fig. 3 | Validation of top miR-132-upregulating candidate compounds in primary rat cortical neurons and human NGN2-iNs. a** Top compounds that showed significant upregulation of miR-132 in primary rat cortical neurons and human NGN2-iNs after 24 h treatment (RT-qPCR analysis, unpaired two-tailed Student's *t* test compared to DMSO controls, $N = 8$ biological replicates for DMSO and forskolin, $N = 4$ for others). **b** Dose curve experiments were performed in DIV14 rat primary neurons after 24 h treatment. Solid lines were used for cardiac glycosides, and dotted lines were used for other compounds. $EC_{50}$ and max fold change were calculated using sigmoidal fit, 4 parameters. ($N = 3-8$ biological replicates). **c** Cardiac glycosides, forskolin, and BIX02188 upregulated the primary transcript of

miR-132 24 h after treatment in rat primary neurons (unpaired two-tailed Student's *t* test compared to DMSO control, $N = 4$ biological replicates). **d**, **e** Upregulation of miR-132 in rat primary neurons was completely blocked by pretreatment with CREB inhibitor ($N = 3$ biological replicates) or actinomycin D ($N = 4$ biological replicates, one-way ANOVA, followed by Dunnett's multiple comparisons test comparing to DMSO/DMSO control). **f** Knocking-down ATP1A1 or ATP1A3, the predominant isoforms in neurons, upregulated pri-miR-132 and mature miR-132 in rat primary neurons (unpaired two-tailed Student's *t* test compared to DMSO control, $N = 3-4$ biological replicates for ATP1A1 KD, $N = 6$ for ATP1A3 KD). All error bars represent mean +/− SD. Source data are provided as a Source Data file.

expression of the primary transcripts and the mature forms of miR-132 and miR-212 overtime (Fig. 4a, b). Both pri-miR-132 and pri-miR-212 were rapidly upregulated following treatment, peaked at 8 h, and rapidly declined to baseline after 72 h (Fig. 4a). In comparison, mature miR-132 and -212 were upregulated at slower kinetics, peaked at 24 h, then slowly declined but were still ~2-fold above baseline at 72 h (Fig. 4b). We speculated that the increase in miR-132 expression would lead to the downregulation of its targets. Indeed, we observed a time-dependent downregulation of *MAPT*, *FOXO3a*, and *EP300* mRNAs that matched the upregulation of miR-132 (Fig. 4c). mRNA targets were

significantly reduced to ~50% of baseline at 24 h and to ~75% of baseline at 72 h, which was similar to the observed effects for miR-132 mimics 72 h after transfection (Supplementary Fig. S6A–C), suggesting that effects of cardiac glycosides are at least partially via miR-132 upregulation. Tau, pTau S202/T305 (AT8), pTau S396, and FOXO3a proteins were also downregulated, though the ratio of pTau: total Tau was unchanged (Fig. 4d–h). In primary PS19 mouse neurons that overexpress human mutant Tau P301S[43], oleandrin upregulated miR-132 and downregulated both mouse *MAPT* and human *MAPT* after 72 h treatment (Supplementary Fig. S6D–H).

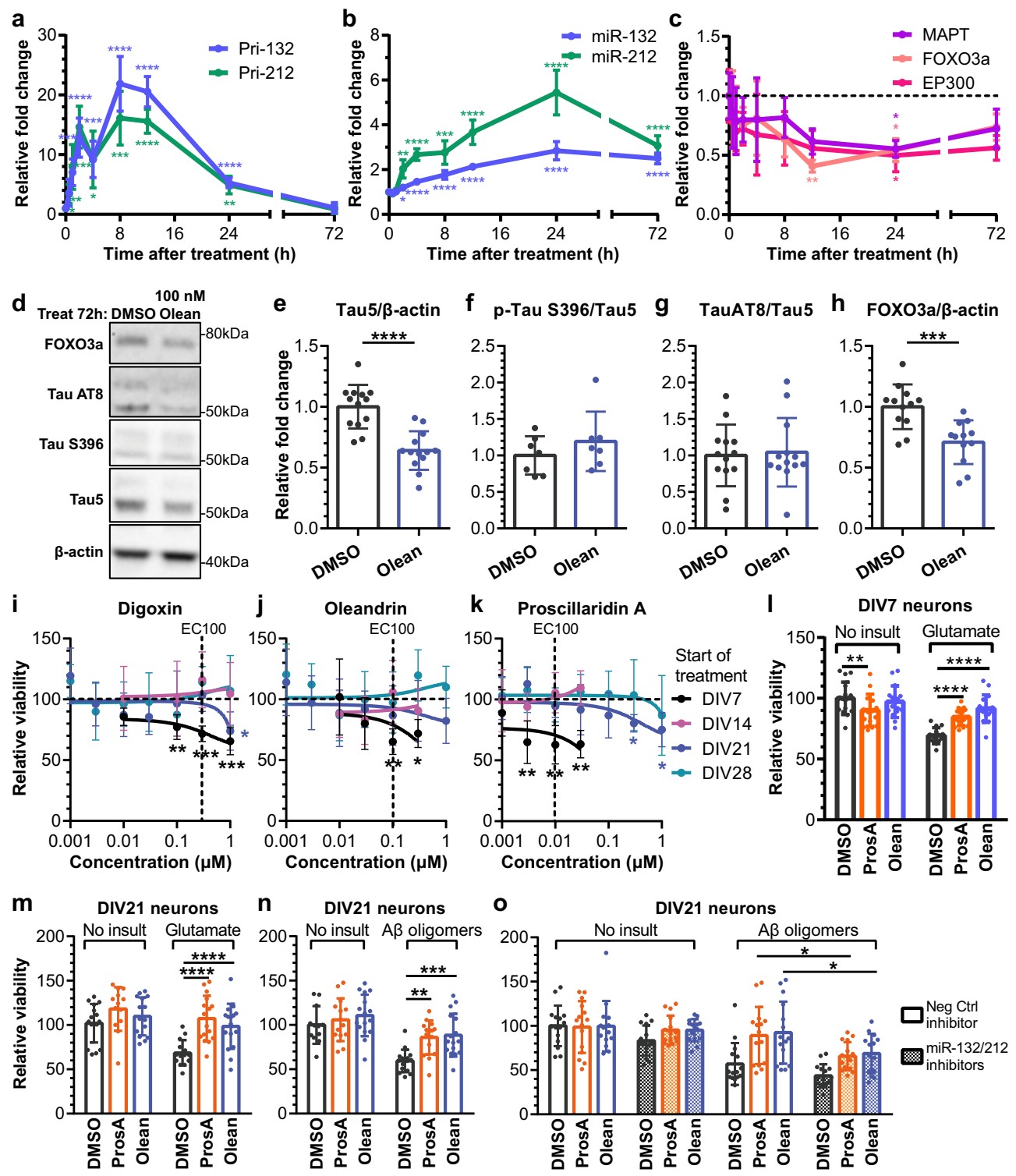

We further hypothesized that the upregulation of miR-132 by the cardiac glycosides would be neuroprotective against various disease-related stress insults, such as excitotoxic glutamate or Aβ oligomers[25]. As several studies have reported possible neurotoxic effects associated with cardiac glycosides[44,45], we first treated rat neurons at different ages in vitro (DIVs 7/14/21/28) with digoxin, oleandrin, and proscillaridin A for 96 h before measuring cellular viability. Interestingly, DIV7 neurons were highly susceptible to cardiac glycoside toxicity, with significant loss of viability observed at the miR-132 $EC_{100}$ for all compounds tested (Fig. 4i–k). However, mature neurons were more resistant to cardiac glycoside toxicity, and no loss of viability was

observed at miR-132 $EC_{100}$ for neurons treated at DIV14 or later (Fig. 4i–k).

As we previously showed that miR-132 mimics rescued loss of viability in younger neurons treated with glutamate[25], we first treated DI7 rat neurons with oleandrin and proscillaridin A at $EC_{100}$ for 24 h, followed by 100 μM glutamate. We observed a slight loss of viability due to proscillaridin A at baseline (Fig. 4l). However, oleandrin and proscillaridin A rescued loss of viability caused by glutamate excitotoxicity (Fig. 4l). Oleandrin and proscillaridin A also partially and dose-dependently rescued neurite loss induced by glutamate without affecting neurite at baseline (Supplementary Fig. S7). Next, we treated

**Fig. 4 | Cardiac glycosides upregulate miR-132 to downregulate miR-132 targets and provide neuroprotection in primary rat cortical neurons. a–c** 100 nM oleandrin upregulated pri- and mature miR-132/212 and downregulated their mRNA targets over time (RT-qPCR analysis, $N \sim 6$ biological replicates, unpaired two-tailed t-test comparing to 0 h, adjusted for multiple comparisons). **d–h** Oleandrin downregulated total Tau ($N = 13$ biological replicates), pTau (AT8, $N = 13$, and S396, $N = 7$), and FOXO3a ($N = 12$) protein after 72 h treatment (Western blot analysis, unpaired two-tailed Student's t test). **i–k** Less mature neurons were more susceptible to cardiac glycoside toxicity, whereas more mature neurons were resistant. Primary rat neurons were treated with various doses of digoxin, oleandrin, and proscillaridin A for 96 h before viability was measured using WST-1. Cells treated at DIV7 showed a dose-dependent reduction in viability. In contrast, cells treated at DIV14, 21, or 28 showed little loss of viability, particularly at EC100 for miR-132 upregulation (unpaired t-test comparing to DMSO, adjusted for multiple comparisons, $N = 4–10$ biological replicates per dose). **l** For DIV7 neurons, proscillaridin A was mildly toxic at baseline. However, both proscillaridin A and oleandrin fully rescued viability loss due to glutamate treatment (2-way ANOVA, followed by Dunnet's multiple comparisons test, $N \sim 16$ biological replicates per condition). **m, n** For DIV21 neurons, proscillaridin A and oleandrin were not toxic at baseline and rescued viability loss due to glutamate or Aβ oligomer treatment (2-way ANOVA, followed by Dunnet's multiple comparisons test, $N \sim 16$ biological replicates per condition). **o** Pre-transfection with miR-132 and -212 inhibitors partially reduced the rescue of viability provided by proscillaridin A and oleandrin (2-way ANOVA, followed by Tukey's multiple comparisons test, $N \sim 15$ biological replicates per condition). All error bars represent mean +/− SD. Source data are provided as a Source Data file.

DIV21 rat neurons with oleandrin and proscillaridin A at $EC_{100}$ for 24 h, followed by 100 μM glutamate or 10 μM Aβ42. Proscillaridin A and oleandrin pretreatment rescued neuronal viability 72 h after toxic insults without affecting viability at baseline (Fig. 4m, n). To determine if the rescue of viability was due to miR-132 and miR-212 upregulation, we transfected DIV21 neurons with miR-132 and miR-212 inhibitors (50 nM each) or CTRL inhibitor (100 nM) 2 h before cardiac glycoside treatment, and then Aβ42 insults 24 h later. Pretreatment with miR-132 and miR-212 inhibitors partially reduced the rescue of viability (Fig. 4o). This observation suggested that cardiac glycosides partially, but not completely, exert neuroprotection through upregulating miR-132 and miR-212.

## Cardiac glycosides significantly reduce Tau and pTau in human iPSC-neurons

To investigate the effects of cardiac glycosides in human neurons, we utilized two additional iPSC-derived neural progenitor cell (NPC) lines: MGH-2046-RC1 derived from an individual with frontotemporal dementia (FTD) carrying the autosomal dominant mutation Tau P301L (P301L), and MGH-2069-RC1 derived from a healthy individual directly related to MGH-2046 (WT). When differentiated into neurons (iPSC-neurons) for 6–8 weeks, these NPCs represent well-established models for studying tauopathy phenotypes in patient-specific neuronal cells relative to a healthy control[46–48]. Notably, while the two lines have similar viability at baseline, P301L neurons showed increased sensitivity to stressors such as Aβ oligomers, NMDA, and rotenone[48,49]. We focused on proscillaridin A, which had the lowest EC50; oleandrin, which was reported to be neuroprotective and BBB-penetrant[50–52]; and digoxin, which has been in clinical use for decades.

Since miR-132 regulates Tau metabolism[23] and Tau lowering is a promising therapeutic strategy for ADRD[46], we first investigated the effects of cardiac glycosides on Tau protein levels. All three cardiac glycosides tested, proscillaridin A, digoxin, and oleandrin, strongly and dose-dependently downregulated Tau (Fig. 5a, d, g–r; Supplementary Fig. S8A, D, G, J). The treatment led to a clear reduction in total Tau as seen with the TAU5 antibody, as well as the phosphorylated form of tau, pTau S396, that showed reductions in both monomeric and oligomeric, high MW (>250 kDa) pTau species, particularly in the mutant Tau P301L neurons. For total Tau (TAU5), the upper band (>50 kDa, monomeric Tau + post-translational modifications (PTMs)) was more intense at lower concentrations. With increasing concentrations, the upper band disappeared, whereas the lower band (<50 kDa, possibly non-pTau) became slightly more intense. This downward band shift suggested that proscillaridin A reduced both Tau accumulation and altered PTMs. Consistent with the latter, proscillaridin A reduced the monomeric form of pTau S396 (~50 kDa) as well as the high molecular weight oligomeric pTau S396 (≥250 kDa). Generally, the three cardiac glycosides tested tend to decrease pTau content at lower doses, starting with a reduction in oligomeric species. At higher doses, an overall reduction in monomeric and oligomeric tau was observed.

RT-qPCR was performed on a matched set of WT and P301L iPSC-neurons and showed a reduction in *MAPT* mRNA, a large increase in pri-miR-132, and a smaller increase in mature miR-132 (Fig. 5b, c, e, f). Similar results were obtained with digoxin and oleandrin (Supplementary Fig. S8). In contrast to the saturation curves observed for mature miR-132 upregulation in rat neurons (Fig. 3b), the majority of the dose responses in iPSC-neurons appear to be inverted U-shapes (Fig. 5b, c, e, f; Supplementary Fig. S8F, I, K, L). The inverted-U response may be due to the greater heterogeneity and more limited maturity of WT and P301L neuronal cultures and that at high doses, cardiac glycosides may trigger additional toxic pathways that decrease miR-132 expression levels. Further immunoblot results showed that in P301L iPSC-neurons, 72 h treatment with 1 μM proscillaridin A, digoxin, or oleandrin reduced both soluble and insoluble total Tau and pTau S396 (Supplementary Fig. S9A–C). The treatment also resulted in a dose-dependent reduction in miR-132 targets at the protein levels, including FOXO3a, EP300, GSK3β, and RBFOX1 (Supplementary Fig. S9D–O).

For all compounds, the concentration of 10 μM was associated with >70% reduction in Tau and pTau with 24 h and 72 h treatments. However, this concentration also reduced neuronal synaptic markers, including post-synaptic density protein 95 (PSD95), synapsin 1 (SYN1), and β-III-tubulin representative of microtubules' structural integrity. These results suggest cardiac glycosides can compromise neuronal integrity at high concentrations and prolonged exposure. Nevertheless, for each compound, we observed a significant safety window in which Tau lowering was not associated with reduced synaptic or microtubule markers (Fig. 5g–r). In all graphs, the yellow shade indicates the dose range where the loss of at least 2 synaptic markers was 30% or less. Notably, WT neurons appeared more susceptible to loss of synaptic markers upon treatment than P301L neurons, particularly at 72 h. For example, proscillaridin A was less toxic to P301L neurons than WT neurons (Fig. 5o–r). Whether the difference in sensitivity to cardiac glycosides between WT and Tau P301L neurons can be reproduced in other human iPSC-neuron lines bearing disease-relevant Tau mutations remains to be tested.

## Cardiac glycosides are neuroprotective in human iPSC-neuronal models of tauopathy

To examine the effects of the cardiac glycosides on neuronal viability, WT and P301L iPSC-neurons were treated with various doses of digoxin, oleandrin, and proscillaridin A for 24 h or 72 h. A dose-dependent loss of viability was observed with all three compounds, particularly at 72 h. Tau-WT neurons had up to 30% loss of viability after 72 h treatment, particularly at the highest dose of 10 μM (Fig. 6a, c). Interestingly, in Tau P301L neurons, the toxicity observed was minimal, with <15% viability loss at the highest concentrations at 72 h (Fig. 6d–f). Given the inherent technical variability across culture wells of iPSC-neurons, cultured for >6 weeks of differentiation, which results in viability reads within 5–10% variability across replicates[46,49], and accounting for the standard deviation error within the assay

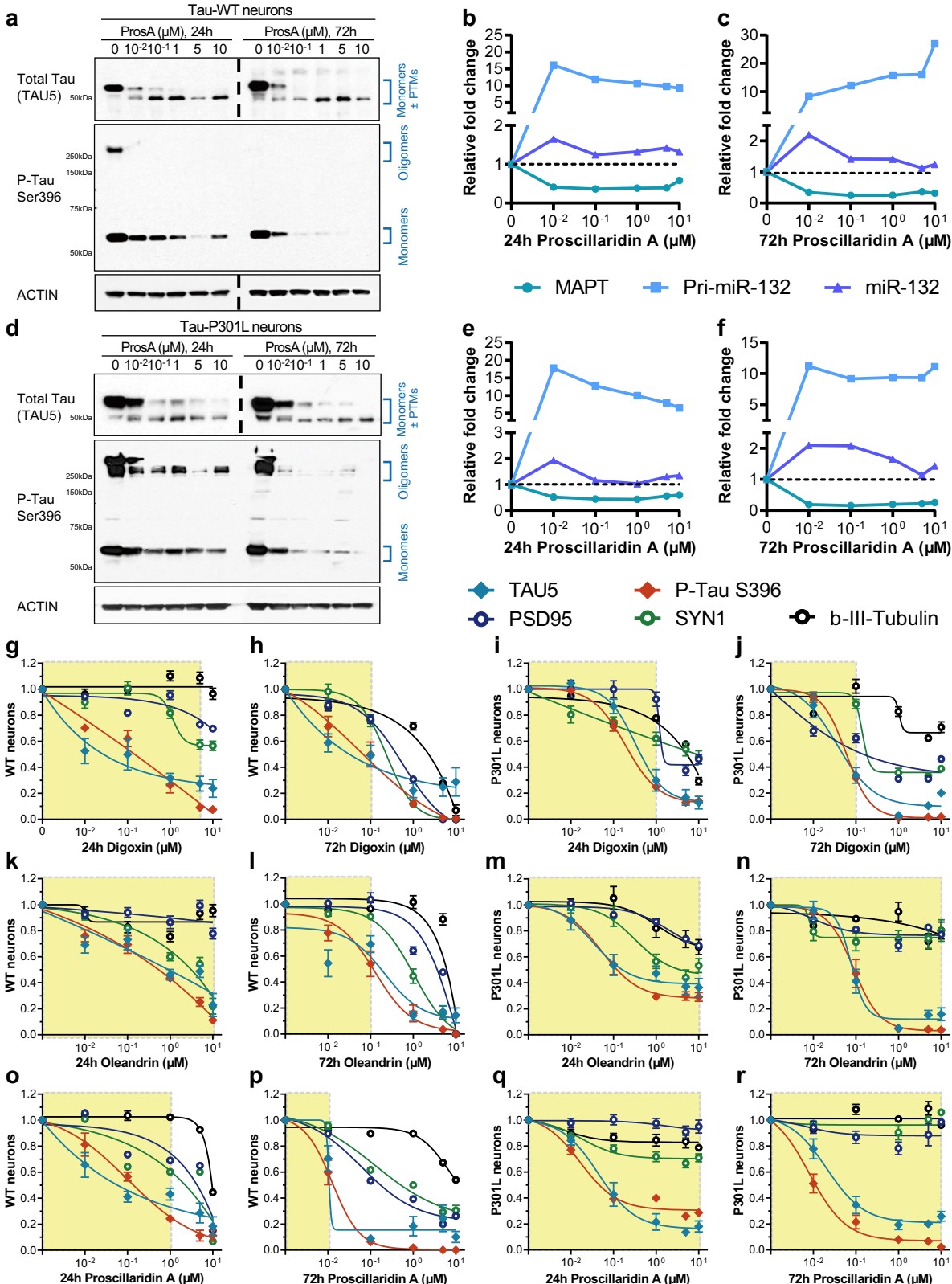

(5–10%), these results suggest a negligible effect on P301L neurons viability at 72 h treatment. These results were consistent with the previous immunoblot data (Fig. 5g–r), showing that P301L neurons were more resistant to cardiac glycoside toxicity than WT neurons.

We next tested whether cardiac glycosides can protect human neurons from various stressors that specifically affect human iPSC-neurons expressing mutant Tau[48]. These include the excitotoxic agonist of glutamatergic receptors NMDA, an inhibitor of the mito-chondrial electron transport chain complex I, rotenone, and the aggregation-prone Aβ42 amyloid peptide. Tau P301L neurons differ-entiated for 8 weeks were pretreated with cardiac glycosides for 6 h prior to adding stressors for 18 h, and viability was measured at the 24 h time point (Fig. 6g). Cardiac glycosides were added at 1 μM and 5 μM, resulting in a marginal decrease in cell viability in P301L neurons

**Fig. 5 | Cardiac glycosides upregulate miR-132 to downregulate miR-132 targets and provide neuroprotection in primary rat cortical neurons.** WT and P301L neurons were differentiated for 6 weeks, then treated with cardiac glycosides for 24 h or 72 h. **a** Representative Western blot for WT neurons treated with proscillaridin A (ProsA). A dose-dependent reduction in total Tau and p-Tau S396 was observed at both 24 h and 72 h. The dotted lines indicated that separate Western blots were put together. Similar results were obtained independently 3 times. **b**, **c** In parallel, a reduction in *MAPT* mRNA and an increase in pri-miR-132 and miR-132 RNA

were observed (*N* = 1 biological sample). Similar results were also observed in Tau P301L neurons by Western blot (**d**) and mRNA (**e**, **f**) analysis. **g**–**r** Western blot densitometry quantification of dose-dependent effects on Tau (TAU5), pTau S396 and the synaptic makers PSD95, SYN1, and β-III-Tubulin (*N* = 3 biological replicates) in WT and P301L neurons treated for 24 h or 72 h. The yellow shades indicate compound concentrations leading to <30% loss of at least two synaptic/microtubule markers. All error bars represent mean +/− SEM. Source data are provided as a Source Data file.

by less than 15% at 24 h (Fig. 6d–f). This reduction is in close proximity to the technical variability typically observed in long-term cultures of iPSC-neurons. All cardiac glycosides significantly rescued neuronal viability in the presence of stressors (Fig. 6h–j). The rescue could also be observed with immunofluorescent staining (Fig. 6k). At baseline, 1 μM of digoxin, oleandrin, or proscillaridin A reduced Tau staining in agreement with the immunoblot data (Fig. 5d, Supplementary Fig. S6) without visibly affecting neuronal health. Treatment with the stressors led to a significant loss of neurites and cell body number in neurons pretreated with vehicle alone and measured by cell viability reduction to 25–60% of control neurons. Neuronal viability loss was significantly rescued to 75–95% of control when neurons were pretreated with the Tau-reducing cardiac glycosides ahead of exposure to stressors such as NMDA, rotenone, or aggregating amyloid-β peptides (Fig. 6h–k). Overall, these results demonstrate that treatment with a low concentration of cardiac glycoside had a neuroprotective effect in human tauopathy neurons exposed to external stressors.

## Transcriptome analysis of human iPSC-neurons confirms shared pathways affected by cardiac glycosides

To further investigate the molecular mechanisms of cardiac glycosides' neuroprotection, we profiled transcriptomes of human iPSC-neurons after 72 h of treatment with increasing doses of digoxin, oleandrin, proscillaridin A or vehicle alone (0.1% DMSO) using RNA sequencing (Fig. 7a). Starting from low doses, cardiac glycosides remarkably changed the global transcriptomes of Tau P301L neurons, as seen in principal component analysis (Fig. 7b), with a single principal component (PC1) being able to clearly separate controls from treatments. More importantly, three different cardiac glycosides regulated transcriptomes similarly and in a prominent dose-dependent manner (Fig. 7b). Differential expression analyses identified thousands of genes significantly regulated with fold-change higher than 4, including many of miR-132 targets based on miRTarBase, TargetScan, or miRDB (Fig. 7c, Supplementary Fig. S10, Supplementary Data 7). The relatively low proportion of miR-132 predicted targets among the down-regulated genes could be due to cascades of downstream regulated genes, as well as multiple pathways affected by the compounds. Many genes were related to neuronal health and activity, including the strongly upregulated *ARC*, which encapsulates RNA to mediate various forms of synaptic plasticity[53,54], and downregulated *MAPT* and the *SLITRK3/4/6* family, which plays a role in suppressing neurite outgrowth[55]. We further focused on the biological pathways that were commonly regulated by all three cardiac glycoside compounds. Notably, these treatments affected many shared pathways (Fig. 7d). Downregulated genes belong to 74 pathways related to neuronal development, morphology, health, or activity (Fig. 7e). Upregulated genes were highly enriched in positive regulators of transcription, negative regulators of programmed cell death, and regulators of stress and unfolded protein response (Fig. 7f). To be noted, pathways enriched for upregulated genes are not necessarily activated since they may include both positive and negative regulators. Among the sub-categories of "regulation of cellular response to stress," "cellular response to starvation" was the most significantly enriched pathway. It could be speculated that cardiac glycosides might affect cellular uptake of nutrients and become toxic when high dosage. In addition,

dozens of transcription factors that had binding sites on *MIR132* promoter and may upregulate its expression, including CREB5, were commonly upregulated by cardiac glycosides (Fig. 7g). The neuro-protective BDNF signaling pathway was significantly upregulated (Supplementary Fig. S8). Therefore, while digoxin, oleandrin, and proscillaridin A all induced miR-132 expression, they likely regulated multiple pathways as well. Overall, shared transcriptomic alterations and regulated pathways further confirmed the common molecular mechanisms of action of cardiac glycosides and their ability to activate stress-protective programs in highly vulnerable Tau-mutant neurons (Fig. 7h).

## Discussion

As miRNAs have been increasingly recognized as master regulators of many biological processes and promising therapeutic targets, screens for miRNA modulators have recently emerged. Several studies have reported successful screens for small molecules that inhibit the activity of specific pathogenic miRNAs, including miR-21[10,13], miR-122[11], and miR-96[15]. Small-molecule inhibitors of miRNAs can be chemically modified to improve pharmacological properties and efficient CNS delivery, though with potentially inferior target specificity relative to miRNA antisense oligonucleotides. Small-molecule inducers of specific miRNAs could provide additional advantages as therapeutics. This is because miRNA supplementation therapies based on oligonucleotides (similar to siRNAs) require chemical modifications for stabilization and durable activity in vivo, which may reduce overall potency in the simultaneous regulation of multiple downstream targets[56]. To date, no miRNA inhibitor or mimic oligonucleotide therapeutics have been approved by the FDA, very few reporter-based screens have been published, and no systematic screens relying on broader miRNome-level readouts have been performed for small-molecule miRNA modulators[14].

Most HTSs for modulators of gene expression employ cell lines as screening platforms and gene-specific heterologous reporter systems as primary assays. However, proliferating, immortalized cells have limited value for identifying neuroprotective agents, and neurons are known to be technically challenging to transfect efficiently and uniformly on a large scale[9]. Here, we applied HTS with miRNA-seq to directly quantify expression levels of hundreds of miRNAs in human neurons treated with small molecule compounds. Many compounds in the screened library have already been approved for clinical usage (*N* = 752) or are currently in phase 2/3/4 clinical trials (*N* = 198) and have been found to be safe for patients (Supplementary Data 1–3). An example is rivastigmine tartrate, which is clinically used for treating mild to moderate dementia caused by Alzheimer's or Parkinson's disease. Notably, the present study is the first HTS-HTS (High-Throughput Screen coupled with High-Throughput Sequencing) for small RNAs that was enabled by the low-input requirement of RealSeq technology[31], though HTS-HTS for mRNA has been conducted previously[57–59]. Some limitations of this pilot screen include its small scale of ~1900 compounds (1370 after quality control), *N* = 1 for each compound, and significant batch effects that required ComBat adjustment. Nevertheless, we successfully validated 4 different compound classes that upregulate miR-132, most notably the cardiac glycosides family. As the first small molecule screen for neuronal miRNA

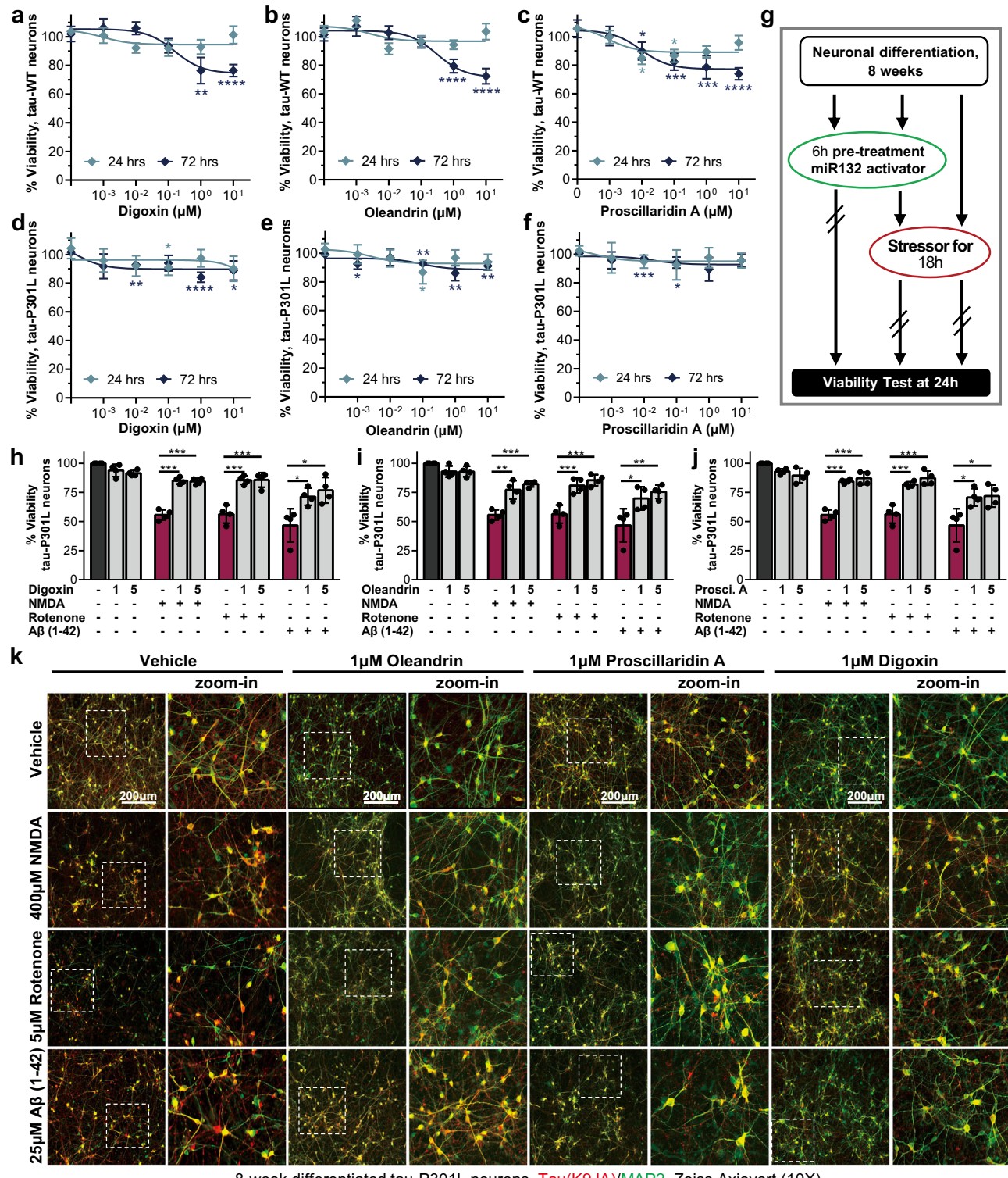

8-week differentiated tau-P301L neurons, Tau(K9JA)/MAP2, Zeiss Axiovert (10X)

**Fig. 6 | Cardiac glycosides protect human Tau P301L iPSC-neurons from diverse toxic insults.** Compounds concentration effect on neuronal viability after 24 h or 72 h treatment of WT (**a**–**c**) and Tau P301L (**d**–**f**) human neurons (Data points indicate mean ± SD; N = 6 biological replicates; unpaired two-tailed Student's t test). **g** Schematic of the assay used to measure neuroprotective effects by cardiac glycosides in tauopathy neurons. **h**–**j** Cardiac glycosides rescued the loss of viability in P301L neurons due to NMDA, rotenone, or Aβ42 oligomer treatment (N = 4

biological replicates, unpaired two-tailed Student's t test). All error bars represent mean +/− SEM. **k** Representative images for P301L neurons at 8 weeks of differentiation treated with cardiac glycosides and each stressor compound. Total Tau (K9JA antibody) staining is shown in red, and MAP2 in green. Similar results were obtained independently 2 times. Scale bars are 200 μm. Source data are provided as a Source Data file.

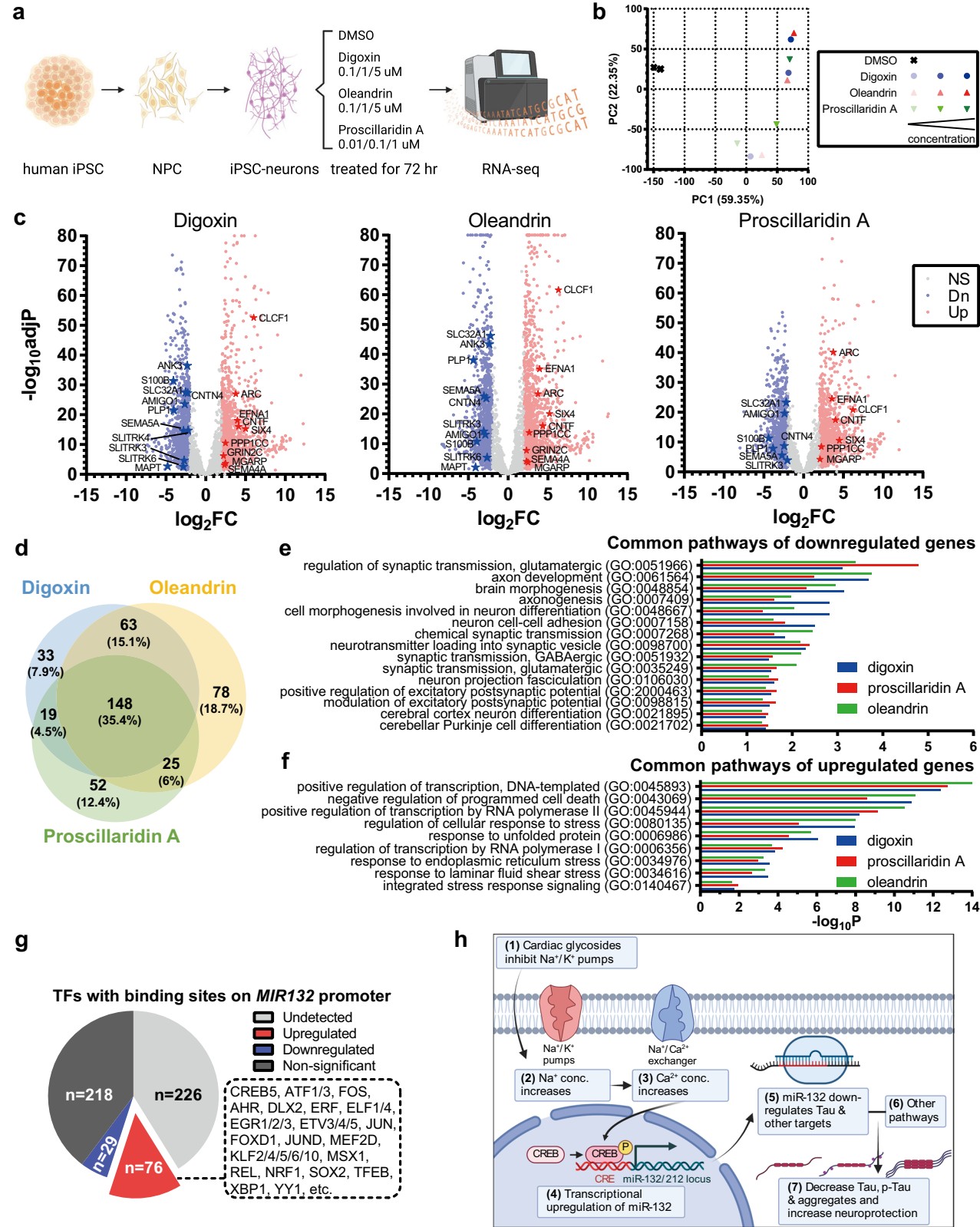

modulators, the dataset can be used to identify compounds that regulate any of the 338 miRNAs, to investigate populations of miRNAs regulated by a class of compounds, to study the relationships between miRNAs, or to unveil compound-miRNA patterns, therefore providing a unique new resource (Supplementary Data 1–3) and facilitating further discoveries of miRNA-regulating mechanisms. As the initial screen was done only once for each drug, the result for any one drug can be

confounded by experimental errors or by the inherent variability in biological systems. Therefore, candidate compounds should be carefully selected and validated. To be noted, the validation rate for miR-132 inducers was comparable to other recently published small compound screens[60–62].

For validation, we focused on miR-132, a master neuroprotector. Several members of the cardiac glycoside family, which are Na+/K+

**Fig. 7 | Transcriptome analysis of human iPSC-neurons treated with cardiac glycosides. a** Workflow of the experiment design (created with BioRender.com). **b** Principal component analysis (PCA) indicated the strong and dose-dependent alteration of global transcriptomic profiles after treatments. **c** Volcano plots showed significant down- and upregulated genes, labeled in blue and red dots, respectively. All dosages were grouped to be compared with DMSO ($N = 3$ vs 3). Stars highlighted dysregulated genes involved in neuronal activity and health. Wald test with FDR correction analyzed using DESeq2. **d** Venn diagram indicated the similarity of pathways affected by three cardiac glycosides. **e** Selected neuronal pathways enriched for the down-regulated genes. **f** Selected transcription- and response-related pathways enriched for the upregulated genes. **g** Effects of cardiac glycosides on the expression of transcription factors (TFs) that have binding sites on *MIR132* promoter. **h** Working model showing the effects of cardiac glycosides: cardiac glycosides act through their conventional mechanism leading to the transcriptional upregulation of miR-132. The increase in miR-132, together with other pathways altered by cardiac glycosides, downregulated various forms of Tau and provided neuroprotection against toxic insults (created with BioRender.com).

ATPase pump inhibitors, were successfully validated to upregulate miR-132 and miR-212 consistently. While we focus on miR-132 due to its great abundance (Supplementary Data 1[41]), many of the effects observed may also be facilitated by miR-212 upregulation. Of note, cardiac glycosides, such as digoxin and digitoxin, are widely used for treating congestive heart failure and cardiac arrhythmias. However, they have a narrow therapeutic index and can be toxic at high doses[63]. Indeed, previously published studies suggested that intraperitoneal injection doses of >1 mg/kg and stereotaxic brain injection of >100 μM in rodent models were associated with adverse effects, including seizures, mania-like behaviors, and death (Supplementary Data 8). However, at low doses (<1 mg/kg and <1 μM), cardiac glycosides were neuroprotective in animal models of stroke[51,64], traumatic brain injury[65], systemic inflammation[66], and ADRD and tauopathies[67,68]. Furthermore, clinical studies suggest that treatment with digoxin might improve cognition in older patients with or without heart failure[69]. Our data supported that the cardiac glycosides reduced Tau accumulation and rescued Tau-mediated toxicity. Further work remains to be done to investigate if any member of the cardiac glycosides can be developed into effective and safe therapeutics for long-term treatment against neurodegenerative diseases. miR-132 EC50 values (Supplementary Data 6) generally correlate with their reported IC50 values[70,71], suggesting that their potency can be further improved with structure-activity relationship enhancement. Oleandrin, previously shown to be neuroprotective with excellent brain penetration and retention[50,51], and proscillaridin A, which exhibited the lowest $EC_{50}$ in rat and human neurons, may be good starting points. PBI-05204, a *Nerium oleander* extract that contains oleandrin as a major active ingredient, was found to be well-tolerated in cancer patients for up to 0.2255 mg/kg/day over 21 consecutive days in a phase 1 clinical trial[72]. Furthermore, as the expression of ATP1A3 is restricted to neurons, whereas ATP1A1 and ATP1A2 are more ubiquitously expressed[73], compounds that selectively target the ATP1A3 isoform may alleviate the systemic impact of the cardiac glycosides, such as on the cardiac system.

Several topics that emerge from our observations warrant further investigation. First, there is a significant difference in the fold change of mature and pri-miR-132. Pri-miR-132 was upregulated by cardiac glycosides by 10 to 30-fold, whereas mature miR-132 in the same treatment group was upregulated by only 1.5-3-fold (Figs. 3 and 5), suggesting a possible bottleneck in processing pri-miR-132 to mature miR-132. Interestingly, miR-132 is downregulated by -1.5-2.5-fold in various neurodegenerative diseases[16]. Furthermore, in mouse models, a 1.5-fold increase in miR-132 expression was associated with spatial memory acquisition, whereas upregulation of >3-fold inhibited learning[74]. These results suggested that the increase promoted by cardiac glycosides is sufficient to restore physiological miR-132 levels. Second, while dysfunctions in ubiquitous ATP1A1 and neuron-specific ATP1A3 have been predominantly linked to neurodevelopmental disorders[75–77], their role in regulating miR-132 may suggest potential underexplored functions in neurodegenerative diseases. Third, several studies have proposed that cardiac glycosides downregulate MAPT and Tau and provide neuroprotection through other pathways, including increased autophagy[67], alternative splicing of MAPT mRNA[78], increased BDNF[52], and inhibiting reactive astrocytes[68]. Our transcriptomic results support that many neuronal pathways are altered, suggesting that cardiac glycosides can modulate multiple pathways that converge on the downregulation of Tau and increase neuroprotection. While further investigation is needed to determine the contribution of miR-132 upregulation to Tau downregulation and neuroprotection, cardiac glycosides emerge as promising therapeutics for neurological disorders, if they can be improved to reduce systemic toxicity and enhance brain penetrance and retention.

In summary, our pilot HTS-HTS of miRNA regulators on human neurons discovered the cardiac glycoside family as novel miR-132 inducers. These small molecules specifically and transcriptionally upregulated miR-132 by inhibiting the Na$^+$/K$^+$ ATPases and protected rat primary neurons and a human iPSC-derived neuronal model of tauopathy against diverse insults. Our pilot study not only highlights cardiac glycosides as promising treatments for neurodegenerative diseases but also provides a key omics resource for future neuronal miRNA regulator discoveries.

## Methods
The research complies with all relevant ethical regulations. Approval for human-derived iPSCs was obtained under the Massachusetts General Hospital/MGB-approved IRB Protocol (#2010P001611/MGH). Experiments involving animals was carried out in accordance with the recommendations in the U.S. National Institutes of Health Guide for the Care and Use of Laboratory Animals. The protocol was approved by the Institutional Animal Care and Use Committee at Brigham and Women's Hospital.

### Induced neuron differentiation from iPSC
Induced pluripotent stem cell (iPSC) lines were retrieved and differentiated into neurons with NGN2 expression, as previously reported[27]. Briefly, iPSCs were plated in mTeSR1 media at a density of $9.5 × 10^4$ cells/cm$^2$ on Matrigel (Corning #354234)-coated plates. Cells were then transduced with the following virus: pTet-O-NGN2-puro (Addgene #52047): 0.1 μL per $5 × 10^4$ cells; Tet-O-FUW-eGFP (Addgene #30130): 0.05 μL per $5 × 10^4$ cells; Fudelta GW-rtTA (Addgene #19780): 0.11 μL per $5 × 10^4$ cells. Transduced cells were dissociated with Accutase (StemCell Technologies) and plated onto Matrigel-coated plates in mTeSR1 (StemCell Technologies) at $5 × 10^4$/cm$^2$ (Day 0). On day 1, media was changed to KSR media (Knockout DMEM, 15% KOSR, 1x MEM-NEAA, 55 μM beta-mercaptoethanol, 1x GlutaMAX; Gibco) with doxycycline (2 μg/ml, Sigma-Aldrich). Doxycycline was maintained in the media for the remainder of the differentiation. On day 2, the media was changed to 1:1 KSR: N2B media with puromycin (5 μg/ml, Gibco), where N2B was composed of DMEM/F12, 1x GlutaMAX, 1x N2 supplement B (StemCell Technologies) and 0.3% dextrose (Sigma-Aldrich). Puromycin was maintained in the media throughout the differentiation. On day 3, the media was changed to N2B media + 1:100 B-27 supplement (GIBCO) and puromycin (10 μg/ml). From day 4 on, cells were cultured in NBM media (Neurobasal medium, 0.5x MEM-NEAA, 1x GlutaMAX, 0.3% dextrose) + 1:50 B-27 + BDNF, GDNF, CNTF (10 ng/ml each, Peprotech). After day 4, half of the media was replaced by fresh media twice per week. Cells were stocked on day 4 at 1 - $2 × 10^6$ cells in 200 μL freezing media (50% day 4 media + 40% FBS + 10%

DMSO) per cryovial in −80 °C overnight, followed by liquid nitrogen storage. iPSC-derived neurons used for validation experiments were prepared similarly. iPSC lines were generated following review and approval through Brigham and Women's Hospital Institutional Review Board (IBR#2015P001676).

## Preparation of NGN2-iNs for high-throughput screen

Twenty-five 96-well plates (Corning) were coated with Matrigel solution (0.2 mg/mL in DMEM/F12) at 60 μL per well for 1.5 h at 37 °C. Then, the Matrigel solution was completely removed, and 100 μL PBS (Gibco) was added per well using electronic 12-channel pipettes in speed 3 (e12c-pip; Eppendorf). The plates were temporality incubated at 37 °C. Frozen day 4 iPSC-iNs were thawed in 500 μL pre-warmed resuspension media per vial, which was composed of NBM, 1:100 B27, and 1:1000 ROCKi (StemCell Technologies), and were kept in a warm metal bath to facilitate the thawing. Multiple vials were pooled into one 50 mL conical tube, then pre-warmed resuspension media were added drop-wisely to reach the volume of 40 mL. After gently mixing by reverting the tube, viable cell concentration was counted with trypan blue (Bio-Rad). The cells were spun down (220 g, 5 min, room temperature) and resuspended in day 4 media at $1 \times 10^5$ cells/mL. Then, PBS was completely removed from the plates, and 100 μL cell suspension was added per well, using e12c-pip at speed 3. The cells were incubated at 37 °C after shaking the plates for even distribution (day 4). To reduce evaporation during incubation, plates were kept in plastic containers lined with sterile wet paper towels. On day 5, an additional 100 μL pre-warmed day 4 medium was added per well using e12c-pip in speed 1. On days 7/10/14/18, 95 μL conditioned medium was removed, and 100 μL pre-warmed day 4 medium was added per well, both using e12c-pip.

## High-throughput screen in NGN2-iNs

On day 19, half (three 384-well plates) of the Selleck bioactive compound library (N = 1902 compounds) were pin transferred (V&P Scientific) to twelve NGN2-iN plates using the Seiko Compound Transfer Robot at 200 nL per well (final concentration at 10 μM). Positive control (Forskolin) and negative control (DMSO) were also pin transferred to the wells without library compound. On day 20, four 10x photos were taken per well automatically using the ImageXpress Micro Confocal microscope (Molecular Devices). Then, the media in the plates was removed with approximately 20 μL media left, using a 24-channel stainless steel manifold (Drummond #3-000-101) linked with a vacuum at a low speed. With the help of Multidrop™ Combi Reagent Dispenser (Thermo Scientific) and the standard cassette (speed: low), 250 μL ice-cold DPBS (Wisent) was added per well. The DPBS was removed with approximately 20 μL liquid left, using another 24-channel stainless steel manifold linked with the vacuum at a low speed. The residual DPBS was completely removed using a mechanic 12-channel pipette. Next, 45 μL lysing solution was added per well using another Multidrop™ Combi Reagent Dispenser with the small cassette (speed: low), where the lysing solution is composed of single-cell lysis buffer (Takara #635013): 1x RNase Inhibitor Murine (NEB): nuclease-free water (Exiqon) = 19:1:190. Thorough lysis was achieved by shaking the plates on a shaker for 5 min at room temperature. The lysis samples were transferred from four 96-well plates to each 384-square-well plate using the 96-well module-coupled VPrep liquid handler (Agilent) with 30 μL tips (twice without changing tips). After sealing, the 384-square-well plates were spun down at 4000 rpm for 5 min, and 10 μL supernatant was aliquoted to a 384-well plate (Eppendorf) using the 384-well module-coupled VPrep liquid handler. The plates were finally sealed with the PlateLoc Heat Sealer (Agilent) and stored at −20 °C. The other half (three and a quarter 384-well plates) of the compound library were added to the remaining thirteen 96-well plates after one day delay (on day 20), using the identical protocol due to the time consumption. The high-throughput screen was conducted in the ICCB-Longwood Screening Facility, Harvard Medical School.

## Ultra-low input miRNA-seq using the RealSeq

To avoid RNA purification, we used RealSeq-T technology (RealSeq Biosciences) following manufacturer recommendations. In summary, cell lysates were incubated at 70 °C for 5 min on RealSeq hybridization buffer (100 mM NaCl, 50 mM Tris-HCl, 10 mM MgCl2, 1 mM DTT, pH 7.9) with 1x RealSeq biotinylated DNA probes to target all miRNAs in miRbase 21. After 2 h of incubation at 37 °C, 10 μL of RealSeq Beads were added, and miRNAs were captured using a 384-well Magnet Plate (Alpaqua, MA). Following three washes with RealSeq Wash buffer, miRNA was eluted from beads in 10 uL of RNase-free water. All the miRNA elusion was input to prepare sequencing libraries with RealSeq-Biofluids following manufacturer instructions (RealSeq Biosciences). In summary, a single adapter and circularization approach was used[31]. Libraries were barcoded with dual indexes and sequenced with a NextSeq 550 (Illumina, CA). FastQ files were trimmed of adapter sequences using Cutadapt with the following parameters: cutadapt -u 1 -a TGGAATTCTCGGGTGCCAAGG -m 15. Trimmed reads were aligned to the corresponding reference by using Bowtie[79] with the following parameters: bowtie -S --chunkmbs 512 -p 4 -n 1 -l 17 -q -m 25 -k 1 --best --strata. Counts of each miRNA were normalized among samples by total miRNA read counts.

## Bioinformatics for miRNA-seq

Bioinformatics analysis commenced with a count expression matrix of 1437 samples and 2656 miRNAs. Upon preliminary filtering, 2324 miRNAs with low baseline expression (in DMSO samples) were excluded, leaving 338 miRNAs for subsequent analysis. The 338 miRNAs were subjected to Trimmed Mean of M-values (TMM) normalization and ComBat batch correction to remove residuals of batch effects. Batch correction efficacy was assessed by conducting a Principal Component Analysis (PCA) in both pre- and post-ComBat. The batch-corrected miRNA data are provided in Supplementary Data 3.

Uniform Manifold Approximation and Projection (UMAP) was applied, and the Silhouette Method was utilized to ascertain the optimal number of compound clusters. A supervised differential expression analysis using a generalized linear model was performed. The association of miRNA signatures with respective clusters was defined using a false discovery rate (FDR) cut-off of $q < 0.0001$.

## Rat and mouse primary neuron culture

Rat primary cortical neuron cultures were prepared from E18 SAS Sprague Dawley pups (Charles River). Brain tissues were dissected, dissociated enzymatically by 0.25% Trypsin-EDTA (Thermo Fisher Scientific), triturated with fire-polished glass Pasteur pipettes, and passed through a 40 μm cell strainer (Sigma-Aldrich). After counting, neurons were seeded onto poly-D-lysine (Sigma-Aldrich) coated cell culture plates at 80,000 cells/cm² in neurobasal medium supplemented with 1X B27 and 0.25X GlutaMax. Half medium was changed every 4 days until use. Mouse primary cortical neuron cultures were prepared from P1 or P2 postnatal pups from PS19 mouse (B6;C3-Tg(Prnp-MAPT*P301S)PS19Vle/J) breeding pairs. Mice were maintained on a 12:12-h light/dark cycle (7:00 am on/7:00 pm off) with food and water provided ad libitum before experimental procedures. After dissection, mouse brain tissues were kept in Hibernate-A medium (Thermo Fisher Scientific) at 4°C in the dark for ~4 h. After genotyping, brains from pups of the same genotype, either WT or PS19, were pooled together and dissociated enzymatically with papain solution (Worthington). After dissociation, mouse neurons were prepared and cultured similarly to rat neurons.

## siRNA and miRNA mimics transfection

siRNAs and miRNA mimics were purchased from Dharmacon (Horizon Discovery) and dissolved in nuclease-free water to prepare 50 μM stock concentrations. miR-132, miR-212, and CTRL inhibitors were custom synthesized from Regulus Therapeutics and described

previously[20]. Transfection was performed using NeuroMag (OZ Biosciences). For siRNA knockdown, transfection was performed with 50 nM siRNAs on DIV7 and DIV9, and RNA was collected for analysis at DIV11. Transfection of DIV14 neurons with 50 nM miR-132 or CTRL mimics was performed similarly. RNA was collected for analysis 72 h later at DIV17. Transfection of DIV21 neurons with miR-132 and miR-212 inhibitors (50 nM each) or CTRL inhibitor (100 nM) (Regulus Therapeutics)[20] was performed 2 h before cardiac glycoside treatment.

### RNA extraction, cDNA preparation, and RT-qPCR

Total RNA from cells was extracted using the Norgen Total RNA Purification Kit (Norgen Biotek) following the manufacturer's protocol. DNAse1 was applied during RNA extraction to remove genomic DNA. RNA was eluted in nuclease-free water, and the concentration was measured using Nanodrop (Thermo Fisher Scientific).

For miRNA analysis, 50 ng of RNA was reverse transcribed into cDNA using the miRCURY LNA RT kit (Qiagen). RT-qPCR mix was prepared using the miRCURY LNA SYBR Green PCR kit (Qiagen). qPCR was performed using the QuantStudio 7 Flex System. The cycling conditions were 95 °C for 10 min, 50 cycles of 95 °C for 15 s, and 60 °C for 1 min following dissociation analysis. miRNA expression was normalized to miR-103a for human neurons, and the geometric mean of miR-103a and let-7a for primary rat neurons. For mRNA analysis, 250–1000 ng of RNA was reverse transcribed into cDNA using the High Capacity cDNA Reverse Transcription Kit (Thermo Fisher Scientific). RT-qPCR mix was prepared using the PowerUp SYBR Green Master Mix (Thermo Fisher Scientific). qPCR was performed using the QuantStudio 7 Flex System. The cycling conditions were 95 °C for 10 min, 50 cycles of 95 °C for 15 s, and 60 °C for 1 min following dissociation analysis. mRNA expression was normalized to the geometric mean of 18S rRNA and GAPDH. Quantification was performed using the delta-delta Ct method. miRNA and mRNA primers used were listed in Supplementary Data 9, 10.

### Transcriptome profile by RNA-seq

After quality control by Agilent 2100 Bioanalyzer, the total RNA was used as input for library preparation by Novogene Co., Ltd, followed by high-throughput sequencing on Illumina HiSeq X with PE150 mode to produce approximately 20 M reads per sample. The reads were quality controlled with FastQC, trimmed with Trimmomatic, aligned with HiSat2 to hg38, and quantified with HTSeq-count using the Galaxy platform. Read counts were processed for differential expression analysis using the R package DEBrowser with DESeq2. Pathway analysis was performed by Enrichr. Promoter binding sites were extracted from JASPAR 2022 TFBS via the UCSC genome browser.

### Western blot analysis

Total protein was extracted using RIPA buffer (Boston Bioproducts) supplemented with Complete, Mini, EDTA-free Protease Inhibitor Cocktail (Millipore Sigma). Protein concentrations were determined using the Micro BCA Protein Assay Kit (Thermo Fisher Scientific). Equal amounts of protein were loaded, and electrophoresis was performed in NuPAGE 4 to 12% gradient Bis-Tris polyacrylamide protein gels (Thermo Fisher Scientific). Proteins were transferred to Immun-Blot PVDF membranes (Bio-Rad) and then blocked with 5% milk in tris-buffered saline with 0.1% Tween (TBS-T, Boston Bioproduct) for 1 h. Membranes were incubated overnight with primary antibodies at 4 °C. Blots were washed and incubated with secondary antibodies for 2 h at room temperature. After washing, bands were visualized with ECL chemiluminescence reagents (Genesee Scientific) using the iBright Imaging System (Thermo Fisher Scientific). Band intensity was measured using the Image Studio Lite software (LI-COR Biosciences). Protein expression level was normalized to β-actin or total Tau (Tau5) as appropriate. Primary antibodies were used at 1:1000 dilution and secondary antibodies were used at 1:10,000 dilution. Antibodies and other key resources are listed in Supplementary Data 11.

### WST-1 assay and neurite length measurement

Cell viability was measured by WST-1 reduction assay (Sigma-Aldrich). For the assay, all medium was removed and replaced with 1X WST-1 reagent dissolved in complete neurobasal medium, followed by 3 h of incubation at 37 °C. The absorbance of the culture medium was measured with a microplate reader at test and reference wavelengths of 450 nm and 630 nm, respectively.

Live cell imaging was performed using the IncuCyte™ Live-Cell Imaging System (Essen BioScience). Cell confluency, cell body number, neurite length, and branching points were monitored and quantified using the IncuCyte™ software.

### Human iPSC-neurons from NPC lines

Approval for work with human subjects and derived iPSCs was obtained under the Massachusetts General Hospital/MGB-approved IRB Protocol (#2010P001611/MGH). The NPC line MGH-2046-RC1 (P301L) was derived from a female individual in her 50 s with FTD carrying the autosomal dominant mutation P301L (c.C1907T NCBI NM_001123066, rs63751273). The NPC line MGH-2069-RC1 (WT) was derived from a related female individual in her 40 s carrying the unaffected WT Tau. Fibroblasts from the two individuals were reprogrammed into iPSCs, converted into cortical-enriched neural progenitor cells (NPCs), and differentiated into neuronal cells over 8 weeks by growth factor withdrawal, as previously described[49].

### iPSC-neurons compound treatment for Western blot analysis and semi-quantitative analysis

NPCs were plated at an average density of 90,000 cells/cm² of six-well plates or 96-well plates coated with poly-ornithine and laminin (POL) in DMEM/F12-B27 media and differentiated for 6 weeks. Compound treatment was performed by removing half-volume of neuronal-conditioned media from each well and adding half-volume of new media pre-mixed with the compound at 2X final concentration, followed by incubation at 37 °C. After 24 h or 72 h, neurons were washed in PBS, collected, and lysed. Western blot and densitometry quantifications were performed as previously described[48].

### Tau protein solubility analysis

Neuronal cell lysates and fractionation were prepared based on protein differential solubility to detergents Triton-X100 and SDS, as previously described[80]. Briefly, cell pellets corresponding to ~800,000 cells were lysed in 1% (v/v) Triton-X100 buffer (Fisher Scientific) in DPBS supplemented with 1% (v/v) Halt Protease/Phosphatase inhibitors (Thermo Fisher Scientific), 1:5000 Benzonase (Sigma) and 10 mM DTT (New England BioLabs). Lysates were centrifuged at 14,000 g for 10 min at 4 °C. The supernatants containing Trion-soluble proteins (S fractions) were transferred to new tubes for Western blot analysis. The pellets were resuspended in 5% (v/v) SDS (Sigma) in RIPA buffer supplemented with 1% (v/v) Halt Protease/Phosphatase inhibitors (Thermo Fisher Scientific), 1:5000 Benzonase (Sigma) and 10 mM DTT (New England BioLabs), and centrifuged at 20,000 g for 2 min at room temperature. These supernatants contained proteins of lower solubility/insoluble (P fractions). SDS-PAGE was performed by loading 20 µg of each S-fraction and double the volume of the P-fraction onto pre-cast Tris-Acetate SDS-PAGE (Novex, Invitrogen). Western blots were performed as before. Densitometry quantification (pixel mean intensity in arbitrary units, a.u.) was done with the Histogram function of Adobe Photoshop 2022, normalized to the respective GAPDH intensity in the S-fraction, followed by normalization to Vehicle.

### Neuronal viability assays

For cardiac glycoside's dose-dependent effects on viability, NPCs were plated (~90,000 cells/cm²) and differentiated in 96-well plates for 8 weeks. After treatment with cardiac glycosides, viability was measured with the Alamar Blue HS Cell viability reagent (Life Technologies)

at 1:10 dilution, after 4 h incubation at 37 °C and according to the manufacturer's instructions. Readings were done in the EnVision Multilabel Plate Reader (Perkin Elmer).

For stress vulnerability assays, 1 μM or 5 μM of digoxin, oleandrin, or proscillaridin A was added to the culture media and incubated for 6 h at 37 °C. Then, either 30 μM Aβ42, 5 μM rotenone, 400 μM NMDA, or vehicle (DMSO) alone, was added to each well for an additional 18 h of incubation. At 24 h, viability was measured with the Alamar Blue HS Cell Viability reagent (Life Technologies) and the EnVision Multilabel Plate Reader (Perkin Elmer).

## Immunofluorescence of neuronal cells

NPCs were plated at a starting density of ~90,000 cells/cm² in black, clear flat bottom, POL-coated 96-well plates (Corning) in DMEM/F12-B27 media and differentiated for six weeks, followed by compound treatment. Neurons were fixed with 4% (v/v) formaldehyde-PBS (Tousimis) for 30 min, washed in PBS (Corning), incubated in blocking/permeabilization buffer [10 mg/mL BSA (Sigma), 0.05% (v/v) Tween-20 (Bio-Rad), 2% (v/v) goat serum (Life Technologies), 0.1% Triton X-100 (Bio-Rad), in PBS] for 2 h, and incubated with primary antibodies overnight (Tau K9JA at 1:1000, MAP2 at 1:1000, Hoechst-33342 at 1:2500). Cells were washed with PBS and incubated with the corresponding AlexaFluor-conjugated secondary antibodies at 1:500 dilution (Life Technologies). Image acquisition was done with a Zeiss AxioVert 200 inverted fluorescence microscope.

## Data analysis

Data management and calculations were performed using Prism 9 (GraphPad). Comparisons between two groups were done using the unpaired two-tailed Student's $t$ test. For the comparison of more than two groups, a one-way analysis of variance (ANOVA), followed by post hoc test, was performed. A $P$ value < 0.05 was considered statistically significant, and the following notations are used in all figures: $*P < 0.05$, $**P < 0.01$, $***P < 0.001$, and $****P < 0.0001$. All error bars shown represent mean +/− standard deviation (SD) unless otherwise stated.

## Reporting summary

Further information on research design is available in the Nature Portfolio Reporting Summary linked to this article.

## Data availability

The data supporting the findings of this study are available from the corresponding authors upon request. miRNA-sequencing and mRNA-sequencing data that support the findings of this study were deposited into the Gene Expression Omnibus (GEO) Repository with accession numbers GSE216991 and GSE228348. Source data for Figs. 3–6 and Supplementary Figs. 1, 4–10 are provided in this paper. Source data are provided with this paper.

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

## Acknowledgements

The BWH iPSC NeuroHub provided support for NGN2-iNs related work. The ICCB-Longwood Screening Facility provided the compounds and instruments for performing the high-throughput drug treatment. The NeuroTechnology Studio at Brigham and Women's Hospital provided IncuCyte instrument access and consultation on data acquisition and analysis. Dr. Bradford Dickerson (MGH), Dr. James Gusella (MGH), Diane Lucente (MGH), and Dr. Bruce Miller (UCSF) are thanked for the generous sharing of patient cell lines. Dr. Michelle Arkin (UCSF), Dr. Erik Uhlmann (DFCI), and Dr. Evgeny Deforzh (BWH) are thanked for their helpful discussion, comments, and edits. Ramil Arora and Harini Saravanan are thanked for annotating the compounds, as shown in Supplementary Data 1–3. Dr. Rachid El Fatimy is thanked for the preparation of Aβ oligomers. BioRender was used in the preparation of Figs. 1a and 7a, h, and Supplementary Fig. S11. National Institutes of Health grant R56 AG069127 and the Rainwater Foundation/ Tau Consortium grants awarded to A.M.K. Rainwater Foundation/ Tau Consortium grant awarded to S.J.H. National Institutes of Health grant R01 AG055909 awarded to T.L.Y.P. National Institute of General Medical Sciences grant R44GM115124 awarded to S.B.S.

## Author contributions

A.M.K., Z.W., and L.D.N. conceived and designed the study. Z.W. and L.D.N. equally contributed as first authors. Z.W., L.D.N., J.Z., M.C.S., S.B.S., R.R., C.R.M., J.M.S.S., and C.H. performed experiments and analysis for this study. S.J.H. and A.M.K. provided the resources needed for experiments. T.L.Y.P. provided specific iPSC lines. L.D.N. wrote an original draft of the manuscript. A.M.K., Z.W., and L.D.N. reviewed and edited the manuscript. All authors reviewed and commented on the manuscript.

## Competing interests

S.B.S. and C.H. are employees of RealSeq Biosciences, which performed the RealSeq miRNA-seq. S.J.H. is a consultant/member of the scientific advisory board for Psy Therapeutics, Frequency Therapeutics, Vesigen Therapeutics, 4 M Therapeutics, Souvien Therapeutics, Proximity Therapeutics, and Sensorium Therapeutics, none of which were involved in the present study. Other authors have no competing interests to declare.
