## [Peer Review File · Nature Communications]

Small Molecule Regulators of microRNAs Identified by High-Throughput Screen Coupled with High-Throughput SequencingREVIEWER COMMENTS

Reviewer #1 (Remarks to the Author):

The manuscript entitled 'Small Molecule Regulators of microRNAs Identified by High-Throughput Screen Coupled with High-Throughput Sequencing' by Nguyen et al. offers a novel perspective on how to explore and employ the therapeutic relevance of miRNAs, focusing on candidates with possible therapeutic potential in neurodegeneration, like miR-132. The authors, instead of using oligonucleotide-based or viral approaches to modulate miRNA levels, they employ a high-throughput screen of small compounds, some of which are used in clinical practice and could be repurposed for miRNA targeting in AD-related dementias. Given the importance of multi-targeting approaches to treat complex, multigenic disorders like AD, using miRNAs, or miRNA-inducers, may be of particular relevance.

The manuscript provides an initial basis for exploring further such modalities, however, several points need to be addressed in order to solidify conclusions and interpretations.

Major remarks:

1. Sample size is generally low throughout the manuscript. The authors do mention in the discussion that N=1 for the initial screening or parts of the downstream analyses and assays. In cases where there is/are indeed only one or two replicates, this has to be very clearly discussed in the results section and all pertinent conclusions should be toned down and adjusted accordingly.

2. Similarly, statistics are often missing. Whether this is related to low sample size or not, it has to be very clearly discussed in the text. If no statistics can be applied, then no statements on observed changes can be made. Similarly, if statistics can be applied, but the results yield no significance, this has to be clearly indicated and incorporated into any downstream conclusions/interpretations.

3. The issue of toxicity/safety is another recurrent issue in the manuscript. I will discuss specific points below:

-Where is the information on whether the screened compounds are in clinical use or not indicated? Are the compounds used in all downstream assays also used in clinical trials? Has their safety (in brain) being demonstrated before? This is critical for the validity of the argument in favor of these modalities over oligonucleotides or viruses.

-Oleandrin is used in several assays in the manuscript, however, its toxicity is well known and no clinical evidence exists to prove its safety in the brain.

-The authors assessed the effects of cardiac glycosides on the viability of iPSC-derived WT and P301L neurons and concluded that 'a dose-dependent loss of viability was observed with all three compounds, particularly at 72h'. However, reduced viability was observed already at doses much lower than the 10uM that yielded the most robust effects on Tau and pTau. How reliable are all the downstream observations in the light of increased toxicity?

-The authors claim that P301L neurons were more resistant to cardiac glycoside toxicity compared to the WT line. How do baseline viability levels compare between WT and P301L neurons? Are the results normalized internally within each line (so that starting point of 0 concentration is set at 1 for each line) or are all data normalized to WT neurons' baseline? Is it possible that the baseline viability of P301L neurons is already lower than that of WT cultures? The authors should provide this information and

discuss in the text.

-It is mentioned that 'Cardiac glycosides were added at 1 μ M and 5 μ M, which did not affect cell viability in P301L neurons at 24h'. This is an inaccurate statement, since there is significantly decreased viability at these concentrations at 24h.

-Lines 288-289: The conclusion is not supported by all the data.

4. Are the miRNA screening results corrected for multiple testing? Are the miRNA changes observed in the screen and reported in the manuscript significant?

5. What are the miRNA changes observed in other compound categories indicated in Figs. 2A and B? These results should be included in the manuscript and discussed across categories.

6. The compound prioritization approach is not clear:

-Why were only oleandrin and BIX02188 used to assess the specificity of miR-132 upregulation?

-Why was the pretreatment with ActD only employed for forskolin and oleandrin?

-Why weren't ouabain and digoxin taken along for downstream analysis, since they were shown to affect miR-132 levels in various conditions?

-Why was only oleandrin used to assess miR-132 kinetics in primary rat cortical neurons?

-Why was proscillaridin A selected and included in the rescue glutamate/Abeta experiments in rat neurons (and those thereafter)?

7. The authors should also assess the levels of other miRNAs upon ATP1A1 and ATP1A3 knockdown (to include miRNAs that either remained unchanged or were downregulated in the initial screen).

8. Throughout the manuscript, it is often assumed that the (bulk of) the observed effects of the cardiac glycosides is mediated by miR-132, which is not the case since several other miRNAs and pathways are also affected. This is only discussed later on in the results and the discussion. The text should be adapted in all earlier pertinent cases to reflect that a direct link between miR-132 and the observed glycoside effects cannot be drawn.

9. Often the observations on one or two cardiac glycosides is generalized to reflect the general potential of all modalities of this class. This should be corrected throughout the text. Some more specific examples:

-'Glycosides' is sometimes used in plural, while only 1 of them was tested in a particular assay.

-Line 240: all three glycosides are mentioned as tested, but only data from one are shown.

10. The interpretation of the results of cardiac glycoside treatment in WT Tau and p301L iPSC-derived neurons is not accurate:

-It cannot be claimed that there is a reduction in total Tau; the band profile of Tau reflects not only posttranscriptional modifications, but also different isoforms, hence, quantifications should be shown considering all bands identified by Tau5.

-Oleandrin did reduce also the low-MW band; what do the different results mean? The authors need to accurately discuss these differences.

-The claim that the 'downward band shift suggested that proscillaridin A reduced both Tau accumulation and altered PTMs' is inaccurate based on the above.

11. In WT and P301L iPSC-neurons (Fig. 5), the authors report a dose-dependent reduction in MAPT mRNA upon glycoside treatment, and also a large increase in pre-miR-132, and a smaller increase in mature miR-132. However those are not dose-dependent. The authors should clarify and discuss. How do these results align with those in rat neuronal cultures (Fig. 3B)?

12. Related to the previous remark, the authors state that ‘Similar results were obtained with digoxin and oleandrin’ referring to Fig. S6. Apart from the general issue of unclear or absent reporting of replicates and statistics (if no replicates and/or significance, no claim can be made on any of the observed effects/trends), this interpretation is inaccurate: no change in mature mir-132 in TAU-P301L neurons was observed at 24h upon oleandrin treatment (similarly to the very small change in mature mir-132 in WT neurons upon digoxin treatment at 72h). This raises a crucial question: how is then the downregulation of MAPT explained, since at least under these conditions it cannot be attributed to miR-132-mediated repression (see also remark #8)?

13. In Fig. 6K, images of digoxin-treated cells should be provided; also, images of cells treated with all glycosides at 5uM should be provided. In addition, the provided images show reduced cell numbers in oleandrin- and proscillaridin-treated cells (without stressors) compared to vehicle/vehicle samples, which does not align with the authors’ conclusions. The authors should explain.

14. The whole lists of DEGs derived from the last RNA sequencing experiment should be provided.

15. How do the changes in miRNome and transcriptome upon treatment with the different compounds (common to both screens) correlate to each other? Are there other miRNAs highly correlated with the transcriptomic changes (even if in different systems)?

16. How many of the DEGs were predicted miR-132 targets (among upregulated and downregulated DEGs per treatment)? The authors should analyze and discuss in the context of proposing cardiac glycosides as a therapeutically relevant means to increase miR-132 levels.

17. How do GSK3B, EP300, RBFOX1, CAPN2, FOXO3, TMEM106B change in the RNAseq dataset? The authors should provide these data and discuss.

18. More careful and balanced discussion should be applied on the following observations with respect to the RNAseq dataset:

-Genes involved in neuronal differentiation and neurite outgrowth are downregulated.

-Genes involved in cellular stress (I would assume not only inhibitors as the authors note) are upregulated just upon treatment (without the addition of stressors, at baseline). Any implications on putative toxicity instead of neuroprotection?

19. Fig. S7: A quick re-analysis of the band density in the some of the provided western blots, does not confirm quantifications shown in S7F-O. E.g.:

-Values for Digoxin, 24h, Gsk3b:

1

1,085122

1,249515

0,917028

0,876771

0,871488

-Values for Oleandrin, 72h, Gsk3b:

1

0,920723

0,947185

0,615252

0,761657

0,746346

-Actin levels also decrease with increasing doses, hence normalization to actin yields smaller changes in Gsk3b (and possibly elsewhere as well). The authors should revisit all their calculations and correct resulting plots.

Minor remarks:

1. Supp. Table 5: the numbers of compounds significantly upregulating miR-132 slightly differ from those mentioned in the text (13 & 11 in the table, 12 & 10 in the text).
2. The culture system employed is not always clearly mentioned. This should be indicated in both the text and figure legends.
3. What do data points represent in Fig. S4A-C? If (some of) these are technical replicates, then they should be averaged instead of being plotted separately.
4. The authors should discuss the fact that Tau phosphorylation did not change upon glycoside treatment in rodent neurons in the context of previous (also from their own work) literature supporting a role for miR-132 in Tau (hyper)phosphorylation.
5. Line 284: Reference to Fig. 8 should be reference to Fig. 6.
6. Lines 285-286: digoxin and oleandrin are wrongly indicated as included in Fig. 5D.
7. What is the concentration employed in Fig. 7C?
8. What was the negative control used for the siRNA transfections?
9. Is there a chance that part of the data provided in Fig. S7 are also shown in Figs. 5 and S6?

Reviewer #2 (Remarks to the Author):

This paper describes a synergistic small molecule screening and RNA-seq approach to discover compounds that affect microRNA targets. The paper described known RNA modulatory molecules such as cardiac glycosides as molecules that affect specific microRNAs but esp as is the focus here on miR-132 in tau-diseases. I could also see this approach being used for other compounds

a few comments:

1. targeting RNA herein is a phenotypic read out and therefore it should be explicitly stated. There is a large interest in RNA targeted drug discovery but it can be in two buckets - one is phenotypic screens and the other is targeting the actual RNA and this paper is clearly a phenotypic screen. as an aside, target centric and systematics screens for chemical mater that target RNA precursors has been done before to inhibit them Velagapudi et al, 2014
2. There is no attempt at mechanism here. Is there any effect on the precursor levels of the microRNA132, is the transcription factor or the DNA's ability to be transcribed affected? These simple

studies should be completed. Also, what about the levels of Dicer or Drosha, these are known compounds and some studies on these would be useful. and, explicitly statements about this should be made as it is hard to advance compounds into man without a target and mechanistic rationale.

Phenotypic screens for miR effectors have been done before and this should be described see (Young et al, JACS 2010) and this should be described. Also work of Dieters group on miR-21 phenotypic screens in Angewandte should be described also.

3. It is known that miR-132 is affected in heart failure and is affected by an antagomir, see <https://academic.oup.com/eurheartj/article/42/2/178/5974817>

Have the authors considered phenotype read out via gain and loss of function studies with miR-132 to track the outcomes here

If done in a previous studies, please state here

4. what is the conservation of the target through model organisms?

5. Are there compounds known that target the ATPase that are not cardiac glycosides that can be tested for these effects? I understand the siRNA was done but chemical inhibition and ablation are two different modes of action. Is the protein target responsible for this activity via non-ATPase roles or not? Also, what is the SAR of this effect on the miR versus the SAR for the canonical target. This should be more clearly stated as a weaker ATP-ase targeting compound should have a weaker effect on the miR if these targets overlap.

Overall a good paper where strength is not the compound (is is a messy molecule and there is no new target or any target at all) but the biology and perhaps others could use this approach but the approach by itself is not very different than what has been done in other settings

Reviewer #3 (Remarks to the Author):

Summary

Nguyen et al. conducted a high-throughput chemical screen to identify responsive miRNAs in neurons. Out of several disease-associated miRNAs, the authors focused on miR-132, which was known to be downregulated in ADRD patients. The authors confirmed that several chemicals, including cardiac glycosides, could upregulate pri-miR-132/212 in rodent and human neurons and provide neuroprotective effects. Although the high-throughput small RNA sequencing (RealSeq) coupled with chemical libraries is an exciting and promising strategy, I doubt the impact of this manuscript because of

the lack of impactful novel knowledge and translational value. Therefore, I do not recommend the current form of the manuscript for publishing in Nature Communications.

Major comments

1. One of the possible impacts of this study is to provide a resource to study interactions between miRNAs and small molecules in neurons. However, in the current form of the manuscript, the authors did not thoroughly analyze the whole dataset. For example, a heatmap comparing all miRNAs and all compounds (or compound families) was missing in Fig 2. Therefore, Fig 2A-F looks like cherry-picking examples. The authors should provide an unbiased interpretation of the high-throughput data in Fig 2 before moving on to the miR-132 part. Just putting RNA-seq read counts in Supplementary Table is not sufficient.
2. Because of the batch effect (Fig S2), the authors normalized RPM with DMSO controls in the same batch. However, the authors used the miR-132 plate rank, rather than normalized RPM, to analyze the dataset (Line 161). I wonder whether the normalization method did not work to compare different plates. If the normalization did not work well, how can we interpret Figures 2E and F? Thus, several control plots are needed to support whether the normalization significantly compromised the batch effect. For example, how are the expression level of housekeeping miRNAs (Fig 1D) before and after normalization?
3. Another major issue is the lack of impressive new knowledge. The manuscript starts with a screening of miR-132-modulating chemicals but ends with an ambiguous contribution of miR-132 in the neuroprotective effect of cardiac glycosides. As the authors mentioned, ("...Further investigation is needed to determine the contribution of miR-132 upregulation to Tau downregulation and neuroprotection..."; Line 375), the current form of the manuscript does not clearly interpret the contribution of miR-132 out of the enormous transcriptomic effects (Fig 7) of cardiac glycosides. However, the neuroprotective effect of cardiac glycosides in ADRD and tauopathies was already shown in recent papers (Ref. 56 and 57). The authors may investigate the contribution of miR-132 in the context of cardiac glycosides by including a miR-132 LNA inhibitor in Fig 4L-O. Also, the contribution of miR-132 can be analyzed by grouping differentially expressed genes with and without the miR-132 seed sequence in Fig 7C. However, I think the contribution of miR-132 would be minor based on the huge transcriptomic changes in Fig 7C. If miR-132 has a critical neuroprotective effect, forskolin (Fig 3A) should show a similar result in Fig 4-6.

Minor comments

1. Fig 1E. Show other control (housekeeping) miRNAs on the same plot.
2. Line 143. I don't think 'COX inhibitors GENERALLY downregulated miR-26b'. Only SOME COX inhibitors downregulated miR-26b.
3. Fig 2F, H and Line 153. In many cases, miRNAs with similar sequences have similar read counts because of sequencing errors and mapping problems. I'm not sure whether the reads of miR-376c-3p, miR-376a-3p, and miR-376-b-3p are uniquely counted during analysis. Also, the same issue is raised to miR-103a-3p, miR-103b, and the let-7 family.
4. Fig 2G-H. Is this data from DMSO-treated samples? I couldn't find a sufficient description of the analysis method for these Figures.

5. miR-132-3p and miR-212-3p share the seed sequence (<https://mirgenedb.org/show/hsa/Mir-132-P1>; <https://mirgenedb.org/show/hsa/Mir-132-P2>). Thus, not only miR-132 but also miR-212 may contribute to the target repression.
6. "pre_miR132_R" binds to pri-miR-132 but not to pre-miR-132. Also, "pre_miR212_R" binds to only pri-miR-212. Throughout the manuscript, change pre-miR-132 and pre-miR-212 into pri-miR-132 and pri-miR-212. Also, do not use 'precursor' but 'primary transcript.'
7. Line 192. Briefly explain why the authors used a CREB inhibitor here.
8. Fig 5. The authors utilized only one cell line for WT and another for P301L neurons. I wonder whether the cell viability difference between WT and P301L neurons upon the treatment of cardiac glycosides can be reproduced in other iPSC neurons.

Typo and simple adjustment

1. Fig 2A-D. Include the number of genes per category on the Y axis in the legend.
2. Fig 2E-F. The X-axis labeling does not match the number of data points.
3. Line 71. Ref.22 seems like a wrong reference in this context.
4. Fig 1C, E. Indicate the number of miRNAs and drugs on the X axis.
5. Fig 2E-H. Label the color bar.
6. Line 152. three transcripts  three miRNAs
7. Fig 3B, Fig S3A. It is difficult to interpret because of too much data in a single plot.
8. Line 225. Fig 4I -> 4L

Please note that significant text changes are colored blue in both the response and the revised manuscript.

REVIEWER COMMENTS

REVIEWER #1

The manuscript entitled ‘Small Molecule Regulators of microRNAs Identified by High-Throughput Screen Coupled with High-Throughput Sequencing’ by Nguyen et al. offers a novel perspective on how to explore and employ the therapeutic relevance of miRNAs, focusing on candidates with possible therapeutic potential in neurodegeneration, like miR-132. The authors, instead of using oligonucleotide-based or viral approaches to modulate miRNA levels, they employ a high-throughput screen of small compounds, some of which are used in clinical practice and could be repurposed for miRNA targeting in AD-related dementias. Given the importance of multi-targeting approaches to treat complex, multigenic disorders like AD, using miRNAs, or miRNA-inducers, may be of particular relevance.

The manuscript provides an initial basis for exploring further such modalities, however, several points need to be addressed in order to solidify conclusions and interpretations.

We thank Reviewer 1 for a very thorough inspection of our manuscript and suggestions for a more objective interpretation of our data. In response, we have toned down various claims, cautioned about the small N number for some results, added new data to support that cardiac glycosides act through the same mechanism, added new data and stated earlier on that the effects of cardiac glycosides are likely only partially through miR-132/212 upregulation, and added missing statistics.

1. In cases where there is/are indeed only one or two replicates, this has to be very clearly discussed in the results section and all pertinent conclusions should be toned down and adjusted accordingly.

Due to the expensive nature of high-throughput drug screens, they are often performed with $N = 1$. Examples include Majd et al. Cell Stem Cell 2023 (1120 drugs, 1 validated hit) (1), Gaido et al. Sci Transl Med 2023 (4475 drugs, 1 validated hit) (2), and Liu et al. Nat Commun 2023 (8207 drugs, 1 validated hit) (3). As such, our initial screen was done similarly to recently published research, and our validation rate is similar or better. Nevertheless, we agree that the implications of the small sample size on the result interpretations should be highlighted early on and more frequently throughout the manuscript. We added additional clarifications of when $N = 1$ or $N = 2$ in relevant figure legends. Specific details are included in the response to comment 2.

We revised the result section to clarify that “**Besides the controls, the initial screen was performed with $N=1$ for each compound.**” (Lines 108-109). We further elaborated in the discussion section that: “**As the initial screen was done only once for each drug, the result for any one drug can be confounded by experimental errors or by the inherent variability in biological systems. Therefore, candidate compounds should be carefully selected and validated. To be noted, the validation rate for miR-132 inducers was comparable to other recently published small compound screens (58-60)**” (Lines 406-410).

Regarding the validation experiments involving iPSC-derived neurons, while we entirely agree with the Reviewer that the experimental N-value is of critical importance for accurate conclusions, we would like to call the attention of the Reviewer to three points that make us confident on $N=2$, specifically for the

iPSC-neuronal data. *One*, is the fact that experiments (Fig. 5-6, S8-9) were performed in two independent genotypes with similar results and conclusions. *Two*, the results show effects across multiple concentrations and two time points of treatment, that altogether corroborate the observations. Conclusions were not drawn from any single data point or stand-alone observation. When taking into consideration, 2 genotypes x 2 times of treatment x ≥ 4 effective concentrations for Tau, all done in duplicates and with additional technical replicates for each point, we are certain that the results are well corroborated. In fact, the results do not stand alone but rather re-enforce and validate observations across different systems tested, including NGN-2 neurons and rat primary neurons. *Three*, N=2 refers to two independent neuronal differentiation setups, months apart, but each experiment also included technical replicates (particularly relevant for Fig. 6) represented by $N \geq 3$ independent wells and/or neuronal plates that were tested in parallel but could, in fact, be considered biological replicates.

2. Similarly, statistics are often missing. Whether this is related to low sample size or not, it has to be very clearly discussed in the text. If no statistics can be applied, then no statements on observed changes can be made. Similarly, if statistics can be applied, but the results yield no significance, this has to be clearly indicated and incorporated into any downstream conclusions/interpretations.

We revisited all figures and added statistical analyses or a statement that statistical analyses could not be performed due to the low N number. The changes made are as below:

- Figure 1: add “1902 compounds in total”.
- Figure 2: a new figure with sample sizes labeled.
- Figure 3: (a) add significance marks, and add “unpaired two-tailed Student’s t-test”. Add N numbers.
- Figure 4: (a-c) add “N=4-8”.
- Figure 5: (b-c) add “N=1, technical duplicates of 2 were shown, no statistical test was performed”.
- Figure 6: no change.
- Figure 7: (c) add “N=3 vs 3”.
- Figure S1: (c) add “N=8 vs 4 vs 4, unpaired two-tailed Student’s t-test”; (d-e) add “N=technical duplicates”.
- Figure S2: a new figure with labeled statistics.
- Figure S3: a new figure. No statistics involved.
- Figure S4: no change
- Figure S5: (a) add “N=3”; (c-d) add “N=4”. (e) add “N=6”. (f) add “N=6”.
- Figure S6: (g) add “N=3”.
- Figure S7: add “N=6-8”
- Figure S8: add “N=1, technical replicates of 2 were shown, no statistical tests were performed”
- Figure S9: no change
- Figure S10: a new figure with labeled statistics

Furthermore, source data are provided for all applicable figures, (Figures 3-6, Figures S1, S4-S10) should any reader wants to check or confirm the analyses.

3. The issue of toxicity/safety is another recurrent issue in the manuscript. I will discuss specific points below:

-Where is the information on whether the screened compounds are in clinical use or not indicated? Are the compounds used in all downstream assays also used in clinical trials? Has their safety (in brain) being demonstrated before? This is critical for the validity of the argument in favor of these modalities over oligonucleotides or viruses.

The approval status for clinical usage for compounds in various countries was included in Tables S1-3 under the column “Approval status” (FDA: U.S. Food and Drug Administration; CFDA: China Food and Drug Administration; EMA: European Medicines Agency; PMDA: Japan Pharmaceuticals and Medical Devices Agency; HMA: European Economic Area Heads of Medicines Agencies; DMF: FDA Drug Master Files; NDC: FDA National Drug Code Directory). When available, the most advanced clinical phase of each compound is listed in the column “Clinical phase”. Furthermore, Tables S1-3 also contain the following information: disease indication if approved (“Indication”), molecular targets (“Target”), specific class (“Class”), blood-brain barrier penetrance predicted using admetSAR (“Crosses BBB (predicted using admetSAR)”), and whether compounds were included in SelleckChem CNS library (“Selleck CNS”) or kinase inhibitors library (“Selleck Kinase Inhibitors”).

To highlight the translational relevance, we added to the discussion that “Many compounds in the screened library have already been approved for clinical usage (N=752) or are currently in phase 2/3/4 clinical trials (N=198) and have been found to be safe for patients (Tables S1-S3). An example is rivastigmine tartrate, which is clinically used for treating mild to moderate dementia caused by Alzheimer's or Parkinson's disease.” (Lines 391-395). We are confident that the extensive information provided for each compound will aid researchers in identifying which compounds have the most therapeutic potential for further investigations. The validated hit digoxin is approved by the FDA, DMF, CFDA, NDC, HMA, and PMDA and is on clinical trial (e.g., NCT05014087 and NCT03559868). However, ouabain is not clinically approved or utilized in recent clinical trials.

4. Oleandrin is used in several assays in the manuscript, however, its toxicity is well known and no clinical evidence exists to prove its safety in the brain.

We agree with the Reviewer that cardiac glycosides have known toxicity, especially at high doses that can lead to neurotoxicity and other adverse reactions. However, there are also multiple studies supporting the potential neuroprotective effects of cardiac glycosides at low doses. These studies suggest that cardiac glycosides can be therapeutic, but their narrow therapeutic window is a major limitation. A summary of the effects of cardiac glycosides in rodent models and humans is provided below and included in the revised manuscript (Table S8, partially shown below due to space limitation). We included the correlation between doses and adverse/ benefit effects in the discussion session: “Indeed, previously published studies suggested that intraperitoneal injection doses of >1 mg/kg and stereotaxic brain injection of >100 μ M in rodent models were associated with adverse effects, including seizures, mania-like behaviors, and death (Table S8). However, at low doses (<1 mg/kg and <1 μ M), cardiac glycosides were neuroprotective in animal models of stroke (47, 62), traumatic brain injury (63), systemic inflammation (64), and ADRD and taupathies (65, 66).” (Lines 417-422)

PMID	Compound	Route	Dose	Frequency & duration	System	Effect
32035075	Oleander extract	IP	150 mg/kg	Once	Neurologic	Negative
25400117	Proscillaridin A	IP	7 mg/kg	5 days a week, 3 weeks	Cancer	Positive
29885637	Digoxin	Oral	5 mg/kg	Daily, 7 days	Cardiac	Negative
28300845	Digoxin	IP	2 mg/kg	Daily, 3 days	Retina	Negative

28292827	Oleandrin	IP	3 mg/kg	Once	Cancer	Negative
29899551	Proscillaridin A	IP	3 mg/kg	Daily, 21 days	Cancer	Positive
35809238	Digoxin	IP	0.3 mg/kg	Every other day, 3 weeks	Neurologic	Positive
28292827	Oleandrin	IP	0.3 mg/kg	Daily, 7 days	Cancer	Positive
35064518	Digoxin	IP	0.1 mg/kg	Daily, 21 days	Neurologic	Positive
30594669	Ouabain	IP	1.5 µg/kg	3 times a week, 6 weeks	Neurologic	Positive
25007121	Ouabain	IP	1 µg/kg	3 times a week, 7 weeks	Neurologic	Positive

We chose to focus on oleandrin as our literature survey supported that it was the compound with neuroprotective effects and good blood-brain barrier penetrance (4-6). Notable, PBI-05204, a *Nerium oleander* extract that contains oleandrin as a major active ingredient, was found to be well-tolerated in cancer patients for up to 0.2255 mg/kg/day over 21 consecutive days (7). We included this study in the discussion session as “PBI-05204, a *Nerium oleander* extract that contains oleandrin as a major active ingredient, was found to be well-tolerated in cancer patients for up to 0.2255 mg/kg/day over 21 consecutive days in a phase 1 clinical trial (70).” (Lines 431-433).

5. The authors assessed the effects of cardiac glycosides on the viability of iPSC-derived WT and P301L neurons and concluded that ‘a dose-dependent loss of viability was observed with all three compounds, particularly at 72h’. However, reduced viability was observed already at doses much lower than the 10µM that yielded the most robust effects on Tau and pTau. How reliable are all the downstream observations in the light of increased toxicity?

Characterization of neuronal health upon compound treatment was based on two assays: 1) viability assay with Alamar Blue at 24h and 72h after treatment at all doses tested (Fig. 6A-F); and 2) Western blot analysis of specific neuronal synaptic (SYN1, PSD95) and structural (β -III-tubulin) markers that can reveal modest changes to neuronal integrity (Fig. 5G-R) ahead of reduction in cell viability. Both assays suggest a therapeutic window of doses less than 10 µM where there was significant tau reduction and minimal loss of viability and protein markers of neuronal integrity, therefore supporting the reliability of downstream observations.

6. The authors claim that P301L neurons were more resistant to cardiac glycoside toxicity compared to the WT line. How do baseline viability levels compare between WT and P301L neurons? Are the results normalized internally within each line (so that starting point of 0 concentration is set at 1 for each line) or are all data normalized to WT neurons’ baseline? Is it possible that the baseline viability of P301L neurons is already lower than that of WT cultures? The authors should provide this information and discuss in the text.

The results for the effect of cardiac glycosides on the viability of Tau-WT and Tau-P301L neurons were internally normalized, that is, they are shown relative to each genotype-neuronal population viability when treated with the Vehicle alone (i.e., DMSO = 100% viability). Within the differentiation timing from iPSC-derived neural progenitors (NPCs) used in this study (6 to 8 weeks of differentiation), this and previous studies have not detected a statistically significant, genotype-dependent effect in neuronal viability as measured by the Alamar Blue Assay (not shown). Therefore, this is not a variable included in our results. Importantly, from our work with several patient-derived cell lines (8, 9), we have found small differences in ‘absolute or baseline viability’ between cell lines of less than 15%, a narrow window that is also seen across cultures setup at different time points, and across clonal lines of the same genotype. For this reason and for this study, it seems to us more accurate to present viability % relative to a data point

(in this case, vehicle treatment) within the same cell line. As the Reviewer suggested, we added context that “Notably, while the two lines have similar viability at baseline, P301L neurons showed increased sensitivity to stressors such as A β oligomers, NMDA, and rotenone (46, 47).” (Lines 267 -269).

7. It is mentioned that ‘Cardiac glycosides were added at 1 μ M and 5 μ M, which did not affect cell viability in P301L neurons at 24h’. This is an inaccurate statement since there is significantly decreased viability at these concentrations at 24h.

The Reviewer is correct, and we apologize for the inaccurate statement. We amended the sentence to: “Cardiac glycosides were added at 1 μ M and 5 μ M, which reduced cell viability in P301L neurons by <15% at 24h (Fig. 6D-F)” (Lines 326-328)

8. Lines 288-289: The conclusion is not supported by all the data.

In the revised manuscript, this sentence now reads as indicated below, to be as accurate as possible based on the data presented: “Treatment with the stressors led to a significant loss of neurites and cell body number in neurons pretreated with vehicle alone and measured by cell viability reduction to 25-60% of control neurons. Neuronal viability loss was significantly rescued to 75-95% of control when neurons were pretreated with the Tau-reducing cardiac glycosides ahead of exposure to stressors such as NMDA, rotenone, or aggregating amyloid- β peptides (Fig. 6H-K). Overall, these results demonstrate that treatment with a low concentration of cardiac glycoside had a neuroprotective effect in human tauopathy neurons exposed to external stressors.” (Lines 331-338).

9. Are the miRNA screening results corrected for multiple testing? Are the miRNA changes observed in the screen and reported in the manuscript significant?

In high-throughput screens, N=1 per compound is common given budget constraints. Therefore, researchers frequently use Z-score to prioritize top compounds as hits, e.g., Table S2 in Majd et al. Cell Stem Cell 2023 (1). Here we used the same strategy to choose our hits, and thus included no statistical comparison or related correction for multiple testing. More importantly, we successfully validated two hits and expanded the observation to other cardiac glycosides beyond the screened library. Of note, in the validation experiments, cardiac glycosides showed strong significance even with N=2. All these results ensured the reliability of our screening results.

10. What are the miRNA changes observed in other compound categories indicated in Figs. 2A and B? These results should be included in the manuscript and discussed across categories.

To better analyze the whole miRNome across the whole compound library, also as suggested by Reviewer 3, we have completely re-plotted Figure 2 and described the new results in Lines 125-169. We believe the new version is more informative and valuable to researchers. Due to space limitations, we are unable to include all compound categories in the result or discussion section. However, supplementary tables S1-3 contain the miRNA values and information for all the compounds to allow other researchers to perform analyses on their miRNAs or compounds of interest.

11. The compound prioritization approach is not clear:

-Why were only oleandrin and BIX02188 used to assess the specificity of miR-132 upregulation?

-Why was the pretreatment with ActD only employed for forskolin and oleandrin?

We were able to validate various classes of compounds as miR-132 activators. We subsequently chose to focus on cardiac glycosides and drop other compounds based on the reasons below:

- Forskolin: used as the positive control as it activates the cAMP/CREB pathway, leading to miR-132 upregulation (10, 11) While forskolin is generally safe as an over-the-counter supplement and is purported to be beneficial for treating asthma, heart disease, hypertension, and weight loss (12), there is little clinical evidence to support these benefits. We chose not to focus on forskolin as its pathway for miR-132 activation is known and it is likely to have many mild but nonspecific effects.
- BIX02188: While BIX02188 robustly upregulated miR-132 in rat neurons, it had no consistent effect on miR-132 in human-derived neurons (Fig. S4C-D). As we focus on compounds that can be translated into treatments for human clinical use, BIX02188 was therefore dropped from further consideration.
- Nitazoxanide, a medication for parasite infections, was dropped because of the large EC50 (~10 μM) and the small maximal fold change (~1.5 fold) obtained.
- Various chemotherapeutics that mildly upregulated miR-132, including pelitinib, rigosertib, etoposide, and XL888, were excluded because of observed toxicity in rodent neurons and the small maximal fold change.

Regarding the mechanistic experiments, we initially performed experiments with many compounds (see figure below and new supplementary figure), but subsequently decided to focus on cardiac glycosides, which we believe to have the most potential for translational application. However, some experiments were performed with multiple compounds to give similar results. A dose-dependent effect of CREB inhibitors on miR-132 upregulation is now included as Fig.3D and Fig. S5B and shown below.

We added the justification for focusing on cardiac glycosides: “We focused on cardiac glycosides due to their potency, efficacy, and novelty as miR-132 inducers. Furthermore, as multiple lines of evidence supported that cardiac glycoside acted through the same mechanisms, we selected oleandrin as the representative cardiac glycoside.” (Lines 218-220)

12. -Why weren't ouabain and digoxin taken along for downstream analysis, since they were shown to affect miR-132 levels in various conditions?

-Why was only oleandrin used to assess miR-132 kinetics in primary rat cortical neurons?

Oleandrin was selected as the representative compound as explained in Lines 218-220. However, all 3 compounds were tested in human neurons.

-Why was proscillaridin A selected and included in the rescue glutamate/Abeta experiments in rat neurons (and those thereafter)?

We observed that all 8 cardiac glycosides tested dose-dependently upregulated miR-132 and miR-212. We hypothesized that all 8 cardiac glycosides act through the same mechanism of action, which is the inhibition of the Na/K pump ATPases. We tested combinations of compounds, including combinations of cardiac glycosides, and observed no additional synergistic effects on miR-132/212 level. In contrast, combinations of oleandrin or proscillaridin with forskolin lead to additional synergistic effects, suggesting that cardiac glycosides and forskolin act through non-identical, if possibly overlapping, mechanisms. This led us to conclude that cardiac glycosides act through the same mechanism, and, therefore, that we could focus on “representative” cardiac glycosides. We chose to focus first on oleandrin (most supporting evidence from existing studies for neuroprotective effects and blood-brain barrier penetrance), then proscillaridin A (lowest EC50 of ~3 nM), and then digoxin (FDA-approved). A new Fig. S5E is now included to support that cardiac glycosides likely act through the same mechanism for miR-132 upregulation as shown on the right. We also added to the result section: “We further tested oleandrin and proscillaridin A combinations and observed no additional synergistic effects on miR-132/212 level (Fig. S5E). In contrast, combinations of oleandrin or proscillaridin A with forskolin led to additional synergistic upregulation, suggesting that cardiac glycosides and forskolin act through non-identical, if possibly overlapping, mechanisms.” (Lines 208-212).

13. The authors should also assess the levels of other miRNAs upon ATP1A1 and ATP1A3 knockdown (to include miRNAs that either remained unchanged or were downregulated in the initial screen).

We tested the effects of ATP1A1 and A3 KD on multiple miRNAs previously tested for oleandrin, forskolin, and BIX02188 (Fig. S5A). Only miR-132 and miR-212 were consistently upregulated in both ATP1A1 and ATP1A3 KD conditions (2-ways ANOVA, followed by Dunnet’s multiple comparisons test). This data is now included in the manuscript as Figure S5F.

14. Throughout the manuscript, it is often assumed that the (bulk of) the observed effects of the cardiac glycosides is mediated by miR-132, which is not the case since several other miRNAs and pathways are also affected. This is only discussed later on in the results and the discussion. The text should be adapted in all earlier pertinent cases to reflect that a direct link between miR-132 and the observed glycoside effects cannot be drawn.

To better address this point, we added to the Result section that: “[...] suggesting that effects of cardiac glycosides are at least partially via miR-132 upregulation.” (Lines 231-232). In addition, we performed the viability assay for neurons treated with a combination of miR-132/212 inhibitors and cardiac glycosides. Adding miR-132/212 inhibitors reduced the neuroprotective effect of cardiac glycosides, supporting our claim that the effects were partially mediated by the miR-132/212 pathway. The new panel is now included as Fig. 4O shown below, and the accompanied description “To determine if the rescue of viability was due to miR-132 and miR-212 upregulation, we transfected DIV21 neurons with miR-132 and miR-212 inhibitors (50 nM each) or CTRL inhibitor (100 nM) 2h before cardiac glycoside treatment, and then A β 42 insults 24h later. Pretreatment with miR-132 and miR-212 inhibitors partially reduced the rescue of viability (Fig. 4O). This observation suggested that cardiac glycosides partially, but not completely, exert neuroprotection through upregulating miR-132 and miR-212.” (Lines 254-260). Of note, these experiments on highly sensitive neuronal cultures combine three insults (Ab, cardiac glycoside, and miR-132/ miR-212 oligonucleotide inhibitors), each with a restricted window of activity and some level of toxicity, and thus are quite challenging. Nevertheless, the results support our conclusions.

15. Often the observations on one or two cardiac glycosides is generalized to reflect the general potential of all modalities of this class. This should be corrected throughout the text. Some more specific examples:

- ‘Glycosides’ is sometimes used in plural, while only 1 of them was tested in a particular assay.
- Line 240: all three glycosides are mentioned as tested, but only data from one are shown.

Our evidence suggests that cardiac glycosides act through the same pathway to upregulate miR-132. We consistently observed the same effects for various cardiac glycosides tested, and that no synergistic effect was observed by a combination of cardiac glycosides. Therefore, we are confident that the results observed can be generalized to most cardiac glycosides. For WT and P301L neurons, we revised the text to emphasize that “All three cardiac glycosides tested, proscillaridin A, digoxin, and oleandrin, strongly

and dose-dependently downregulated Tau (Fig. 5A, D; Fig. 5G-R; Fig. S8A, D, G, J). The treatment led to a clear reduction in global Tau as seen with the TAU5 antibody, as well as the phosphorylated form of tau, pTau S396, that showed reductions in both monomeric and oligomeric, high MW (>250 kDa) pTau species, particularly in the mutant Tau P301L neurons.” (Lines 273-278).

16. The interpretation of the results of cardiac glycoside treatment in WT Tau and p301L iPSC-derived neurons is not accurate: -It cannot be claimed that there is a reduction in total Tau; the band profile of Tau reflects not only posttranscriptional modifications, but also different isoforms, hence, quantifications should be shown considering all bands identified by Tau5.

The TAU5 antibody detects total tau protein, and it is quite usual and standard to refer to the tau species recognized by this antibody as “total Tau”, as we did, also in part to distinguish from the species detected by the S396 pTau antibody, which mainly detects phosphorylated Tau at Ser396. TAU5 will detect all forms of tau, including ‘naked tau’ protein, tau with PTMs, and different isoforms endogenously expressed in the neurons. As we know from previous studies from our group and others (8, 13), the main Tau isoform present in these neurons is the fetal form 3R0N with mainly phosphorylation modifications, since phosphatase treatment reduces all bands visible by Western blot to a single band corresponding to 3R0N Tau. Our goal in the present work was to have a readout for total, global Tau independent of PTMs, and a specific readout for phospho-Tau. The clear reduction seen with TAU5 antibody represents an overall reduction in the total Tau, i.e., indiscriminate of isoform or PTMs. Nevertheless, by Western blot alone and without further sample processing or use of other antibodies, we cannot accurately identify the exact Tau species. While we believe that we have used the term correctly as accepted in the field, we have replaced “total tau” with “global tau” in the manuscript to avoid the confusion.

17. Oleandrin did reduce also the low-MW band; what do the different results mean? The authors need to accurately discuss these differences.

If the Reviewer is referring to the lower MW band detected by TAU5, our interpretation is that the compounds can affect Tau (the global Tau detected by TAU5) in two ways: by reducing accumulation of phospho-tau (potentially even reducing tau phosphorylation via other targets such as GSK3), which will lead to the reduction of intensity of the higher band (band downward shift), and/or by reducing the steady-state levels of accumulated total protein that can be visible by the overall reduction in both bands, at and above 50kDa. It is likely that both events are happening and dependent.

The Western blot results for Tau-P301L neurons with the antibody pTau S396 show high MW Tau species of ≥ 250 kDa that are not SDS soluble and did not migrate as other species around the monomeric ~50 kDa size. We know that these high MW species represent oligomers of pTau that are present in patient-derived mutant tau neurons, and we hypothesize that these are the most aggregation prone and toxic species to the neurons in the long term of neurodegeneration etiology. Again, as with TAU5, the cardiac glycosides tend to show changes in pTau content at the lower doses, starting with a reduction in oligomeric species, and then an overall reduction in monomeric + oligomeric tau. This discussion is now included in the revised manuscript in the result section as “Generally, the three cardiac glycosides tested tend to decrease pTau content at lower doses, starting with a reduction in oligomeric species. At higher doses, an overall reduction in monomeric and oligomeric tau was observed.” (Lines 284-286).

18. The claim that the ‘downward band shift suggested that proscillaridin A reduced both Tau accumulation and altered PTMs’ is inaccurate based on the above.

We respectfully disagree with the Reviewer on this point, for the reasons presented in the answer to the previous point, and mainly because we know that the downward shift is due to a PTM and not the reduction of a specific isoform of tau. Study by Sato et al. (2018 Neuron) (13) has shown that the main Tau isoform present in these neurons is the fetal form 3R0N. Another explanation for the downward shift would be Tau cleavage or another PTM (e.g., acetylation or ubiquitination), which we know TAU5 would not be a particularly good readout for (data not shown). So, without mass spectrometry or PTM-specific readouts, and based on the parallel result with pTau S396, changes to phospho-tau seem the most likely. We have also purposefully used the word ‘suggest’ acknowledging that there are other possible but less likely explanations.

19. In WT and P301L iPSC-neurons (Fig. 5), the authors report a dose-dependent reduction in MAPT mRNA upon glycoside treatment, and also a large increase in pre-miR-132, and a smaller increase in mature miR-132. However those are not dose-dependent. The authors should clarify and discuss. How do these results align with those in rat neuronal cultures (Fig. 3B)?

We observed standard saturation dose-response curves for various compounds tested in rat primary neurons and human NGN2-iNs. For WT and P301L neurons differentiated from NPCs, the increase was smaller and generally has an inverted-U response. We described this difference and speculated in the result section that: “In contrast to the saturation curves observed for mature miR-132 upregulation in rat neurons (Fig. 3B), the majority of the dose responses in iPSC-neurons appear to be inverted U-shapes (Fig. 5B, C, E, F; Fig. S8F, I, K, L). The inverted-U response may be due to the greater heterogeneity and more limited maturity of WT and P301L neuronal cultures and that at high doses, cardiac glycosides may trigger additional toxic pathways that decrease miR-132 expression levels.” (Lines 289-294).

20. Related to the previous remark, the authors state that ‘Similar results were obtained with digoxin and oleandrin’ referring to Fig. S6. Apart from the general issue of unclear or absent reporting of replicates and statistics (if no replicates and/or significance, no claim can be made on any of the observed effects/trends), this interpretation is inaccurate: no change in mature mir-132 in TAU-P301L neurons was observed at 24h upon oleandrin treatment (similarly to the very small change in mature mir-132 in WT neurons upon digoxin treatment at 72h). This raises a crucial question: how is then the downregulation of MAPT explained, since at least under these conditions, it cannot be attributed to miR-132-mediated repression (see also remark #8)?

Due to the long time required for the differentiation of WT and P301L neurons (~8 weeks in culture) and high associated costs, we could not obtain additional biological replicates. We have, however, added technical replicates starting from same RNA in the revised Fig. 5 and Fig. S8. The technical replicate reviewed that the DMSO sample used for normalization for Tau-P301L neurons, 24h oleandrin was abnormal, which was confirmed by another technical replicate for these samples only. Overall, 2 cell lines x 3 drugs x 2 time points = 12 conditions showed a ~1.5-2-fold upregulation of mature miR-132. As we previously argued to comment #1, together, the consistency of the results across different drugs, different cell lines, and different times support the conclusions that were drawn.

As previously reported, miR-132 regulates tau metabolism via several targets at multiple levels. A ~2-fold reduction in AD neurons is strongly associated with tau pathology (14). In mouse models, a 1.5-fold increase in miR-132 expression was associated with spatial memory acquisition, whereas upregulation of >3-fold inhibited learning (15). Thus, whereas miR-132 transcription can be strongly induced, the physiological levels of mature miR-132 are tightly controlled in a narrow expression window. miR-132 is one of the most abundant miRNAs in neurons, and its small fold upregulation can be sufficient to

downregulate some of its targets. We also emphasized that the effect of cardiac glycosides on tau and neuronal viability is most likely only partially mediated by miR-132 as already shown in previous responses (see page 8).

21. In Fig. 6K, images of digoxin-treated cells should be provided; also, images of cells treated with all glycosides at 5uM should be provided. In addition, the provided images show reduced cell numbers in oleandrin- and proscillaridin-treated cells (without stressors) compared to vehicle/vehicle samples, which does not align with the authors' conclusions. The authors should explain.

As requested by the Reviewer, we included the digoxin panel in Fig. 6K. Regarding the comment on the “reduced cell number” for some panels, we have done our best to select a representative area of each well of neuronal cultures, and have adjusted accordingly to have a fair representation of the equal number of cells plated, possible variability well-to-well, and compound treatment potential (but statistically insignificant) effect. While there is variability in culturing iPSC-neurons in 96-well plates format for 8 weeks, with media changes and inherent well-to-well heterogeneity, no significant reduction of cell number was observed in oleandrin-, proscillaridin-, and digoxin-treated cultures.

As per the Reviewer's recommendation and answer to the point above, the 5µM dose is already showing some changes to neuronal integrity as measured by the synaptic (SYN1, PSD95) and microtubule (TUJ1) markers by Western blot in Fig. 5G-R. So, consistent with what we now included for Figure 6H-J, we focused the analysis on the 1µM cardiac glycoside concentration, where an effect on Tau and pTau can be measured with insignificant effect on viability or neuronal markers.

22. The whole lists of DEGs derived from the last RNA sequencing experiment should be provided.

The whole list of DEGs from the RNA-seq experiment shown in Figure 7 was provided in Table S7.

23. How do the changes in miRNome and transcriptome upon treatment with the different compounds (common to both screens) correlate to each other? Are there other miRNAs highly correlated with the transcriptomic changes (even if in different systems)?

We agree that such miRNome-transcriptome correlation analysis will be informative for studying the effects of drugs on miRNA activity in neurons. However, it would require an additional high-throughput screen with mRNA sequencing readouts, which is beyond the scope of this study. Nevertheless, we optimized lysing solution for both library preparation technologies (Fig. S1) which will enable such analyses in the future.

24. How many of the DEGs were predicted miR-132 targets (among upregulated and downregulated DEGs per treatment)? The authors should analyze and discuss in the context of proposing cardiac glycosides as a therapeutically relevant means to increase miR-132 levels.

Based on miTarBase, 42, 48 and 40 DEGs were predicted miR-132 targets for digoxin, oleandrin, and proscillaridin A, respectively. Based on TargetScan, 59, 70, and 48 DEGs were predicted miR-132 targets for digoxin, oleandrin, and proscillaridin A, respectively. Based on miRDB, 83, 103 and 71 DEGs were predicted miR-132 targets for digoxin, oleandrin, and proscillaridin A, respectively. We have added corresponding analysis in Page 9, Lines 343-345, as “Differential expression analyses identified thousands of genes significantly regulated with fold-change higher than 4, including dozens of miR-132 targets based on miRTarBase, TargetScan, or miRDB.” (Lines 348-350).

25. How do GSK3B, EP300, RBFOX1, CAPN2, FOXO3, TMEM106B change in the RNAseq dataset? The authors should provide these data and discuss.

The RPMs for these mRNAs, plus ITBPKB, another well-validated (and implicated in Ab metabolism) target of miR-132, were provided below (16). 5 out of 8 mRNA targets showed downregulation. CAPN2 and GSK3B appeared unchanged, whereas EP300 was upregulated. This result is now included as Fig. S10. However, EP300 protein was still downregulated (Fig. S9 D-L), suggesting that miR-132 regulates EP300 mRNA translation, but not its degradation.

26. More careful and balanced discussion should be applied on the following observations with respect to the RNAseq dataset:

- Genes involved in neuronal differentiation and neurite outgrowth are downregulated.
 - Genes involved in cellular stress (I would assume not only inhibitors as the authors note) are upregulated just upon treatment (without the addition of stressors, at baseline). Any implications on putative toxicity instead of neuroprotection?
-

For more balanced discussion and interpretation, we revised the text (Page 9, Line 353-355), as following “To be noted, pathways enriched for upregulated genes are not necessarily activated since they may include both positive and negative regulators. Among the subcategories of "regulation of cellular response to stress," "cellular response to starvation" was the most significantly enriched pathway. It could be speculated that cardiac glycosides might affect cellular uptake of nutrients and become toxic when high dosage.” (Lines 358-362).

27. Fig. S7: A quick re-analysis of the band density in the some of the provided western blots, does not confirm quantifications shown in S7F-O. E.g.:

- Values for Digoxin, 24h, Gsk3b: 1; 1,085122; 1,249515; 0,917028; 0,876771; 0,871488
 - Values for Oleandrin, 72h, Gsk3b: 1; 0,920723; 0,947185; 0,615252; 0,761657; 0,746346
 - Actin levels also decrease with increasing doses, hence normalization to actin yields smaller changes in Gsk3b (and possibly elsewhere as well). The authors should revisit all their calculations and correct resulting plots.
-

We are very sorry that the Reviewer felt the need to quantify densitometry values to make this point. The noted discrepancy highlights that we must change the Figure size and blots images resolution to ensure that the image provided by the journal has better resolution and reflects the quantification made directly

from the X-Ray films (highest resolution). Below are the values used in the graphs. When using full-resolution images, values were normalized to Actin, and then normalized to DMSO, and the values in our original graphs stand as follows below.

Digoxin 24h

X	Group A				Group B				Group C				Group D			
X Title	p300				FOXO3a				GSK3 β				RBFOX1			
X	A:Y1	A:Y2	A:Y3	A:Y4	B:Y1	B:Y2	B:Y3	B:Y4	C:Y1	C:Y2	C:Y3	C:Y4	D:Y1	D:Y2	D:Y3	D:Y4
	1.00	1.00	1.00	1.00	1.00	1.00	1.00	1.00	1.00	1.00	1.00	1.00	1.00	1.00	1.00	1.00
0.010	0.78	0.73	1.03	0.95	0.90	0.88	1.10	1.29	1.11	1.16	0.89	0.83	1.10	1.10	0.91	0.97
0.100	0.81	0.76	1.07	0.99	0.63	0.57	0.76	0.86	1.18	1.19	0.95	0.86	0.93	0.93	0.77	0.81
1.000	0.10	0.03	0.13	0.04	0.44	0.35	0.59	0.72	0.71	0.77	0.57	0.55	0.75	0.74	0.62	0.65
5.000	0.08	0.03	0.11	0.04	0.34	0.24	0.58	0.49	0.72	0.69	0.58	0.50	0.65	0.63	0.54	0.55
10.000	0.08	0.03	0.11	0.04	0.20	0.12	0.40	0.25	0.63	0.61	0.51	0.44	0.44	0.44	0.36	0.39

Digoxin 72h

X Title	p300				FOXO3a				GSK3 β				RBFOX1			
X	A:Y1	A:Y2	A:Y3	A:Y4	B:Y1	B:Y2	B:Y3	B:Y4	C:Y1	C:Y2	C:Y3	C:Y4	D:Y1	D:Y2	D:Y3	D:Y4
	1.00	1.00	1.00	1.00	1.00	1.00	1.00	1.00	1.00	1.00	1.00	1.00	1.00	1.00	1.00	1.00
0.010	0.63	0.51	0.84	0.67	0.45	0.37	0.91	0.76	0.61	0.57	0.49	0.41	1.00	0.84	0.83	0.74
0.100	0.07	0.03	0.09	0.04	0.11	0.03	0.22	0.06	0.52	0.51	0.42	0.37	0.61	0.57	0.51	0.50
1.000	0.12	0.07	0.16	0.09	0.05	0.00	0.10	0.00	0.31	0.26	0.25	0.19	0.68	0.60	0.56	0.52
5.000	0.14	0.07	0.19	0.09	0.04	0.00	0.07	0.00	0.38	0.32	0.30	0.23	0.97	0.83	0.80	0.73
10.000	0.07	0.02	0.09	0.03	0.02	0.00	0.04	0.00	0.51	0.50	0.41	0.36	0.99	0.82	0.82	0.72

Oleandrin 24h

X Title	p300				FOXO3a				GSK3 β				RBFOX1			
X	A:Y1	A:Y2	A:Y3	A:Y4	B:Y1	B:Y2	B:Y3	B:Y4	C:Y1	C:Y2	C:Y3	C:Y4	D:Y1	D:Y2	D:Y3	D:Y4
	1.00	1.00	1.00	1.00	1.00	1.00	1.00	1.00	1.00	1.00	1.00	1.00	1.00	1.00	1.00	1.00
0.010	0.80	0.81	0.85	0.79	1.04	1.02	1.09	1.20	0.98	0.98	1.00	1.07	0.92	0.96	0.84	0.98
0.100	0.68	0.69	0.72	0.67	0.73	0.79	0.76	0.92	0.81	0.82	0.84	0.89	0.91	0.94	0.83	0.95
1.000	0.26	0.25	0.27	0.25	0.58	0.61	0.60	0.72	0.56	0.56	0.58	0.61	0.80	0.80	0.73	0.82
5.000	0.67	0.64	0.71	0.62	0.64	0.61	0.67	0.71	0.49	0.49	0.51	0.54	0.64	0.65	0.58	0.66
10.000	0.51	0.52	0.54	0.51	0.43	0.47	0.44	0.55	0.37	0.38	0.38	0.42	0.61	0.62	0.55	0.63

Oleandrin 72h

X Title	p300				FOXO3a				GSK3 β				RBFOX1			
X	A:Y1	A:Y2	A:Y3	A:Y4	B:Y1	B:Y2	B:Y3	B:Y4	C:Y1	C:Y2	C:Y3	C:Y4	D:Y1	D:Y2	D:Y3	D:Y4
	1.00	1.00	1.00	1.00	1.00	1.00	1.00	1.00	1.00	1.00	1.00	1.00	1.00	1.00	1.00	1.00
0.010	1.07	1.06	1.13	1.03	1.18	0.96	1.23	1.12	1.00	0.86	1.03	0.94	1.11	0.89	1.01	0.90
0.100	0.14	0.15	0.15	0.15	0.33	0.24	0.34	0.29	0.60	0.56	0.62	0.62	0.87	0.74	0.79	0.75
1.000	0.14	0.14	0.14	0.14	0.18	0.15	0.18	0.18	0.26	0.24	0.27	0.26	0.80	0.67	0.72	0.68
5.000	0.01	0.01	0.01	0.01	0.13	0.11	0.13	0.12	0.33	0.29	0.34	0.32	0.78	0.62	0.71	0.63
10.000	0.04	0.04	0.04	0.04	0.09	0.08	0.09	0.09	0.43	0.41	0.44	0.45	0.97	0.85	0.88	0.86

ProsA 24h

X Title	p300				FOXO3a				GSK3 β				RBFOX1			
X	A:Y1	A:Y2	A:Y3	A:Y4	B:Y1	B:Y2	B:Y3	B:Y4	C:Y1	C:Y2	C:Y3	C:Y4	D:Y1	D:Y2	D:Y3	D:Y4
	1.00	1.00	1.00	1.00	1.00	1.00	1.000	1.00	1.00	1.00	1.00	1.00	1.00	1.00	1.00	1.00
0.010	0.95	0.99	1.12	1.15	0.70	0.83	0.640	0.93	1.04	1.02	0.82	0.77	0.85	0.75	0.84	0.72
0.100	0.31	0.30	0.36	0.35	0.52	0.52	0.480	0.67	0.89	0.88	0.71	0.67	1.00	1.03	0.99	0.98
1.000	0.32	0.31	0.38	0.37	0.37	0.33	0.135	0.57	0.83	0.81	0.66	0.61	0.80	0.81	0.79	0.77
5.000	0.00	0.00	0.00	0.00	0.66	0.46	0.580	0.56	0.84	0.83	0.67	0.62	0.71	0.72	0.70	0.68
10.000	0.24	0.23	0.28	0.27	0.01	0.32	0.010	0.38	0.93	0.94	0.74	0.71	0.68	0.69	0.67	0.66

ProsA 72h

X Title	p300				FOXO3a				GSK3 β				RBFOX1			
X	A:Y1	A:Y2	A:Y3	A:Y4	B:Y1	B:Y2	B:Y3	B:Y4	C:Y1	C:Y2	C:Y3	C:Y4	D:Y1	D:Y2	D:Y3	D:Y4
	0.85	0.86	1.00	1.00	1.14	0.82	1.00	1.00	1.00	1.00	1.00	1.00	1.01	1.05	1.00	1.00
0.010	0.70	0.43	0.57	0.67	0.68	0.44	0.59	0.53	1.00	1.05	1.18	1.06	0.99	0.91	0.97	0.87
0.100	0.39	0.21	0.48	0.22	0.30	0.20	0.27	0.24	0.96	0.97	0.76	0.73	0.66	0.67	0.65	0.64
1.000	0.47	0.21	0.55	0.25	0.17	0.09	0.15	0.11	0.88	0.87	0.70	0.65	0.56	0.56	0.55	0.53
5.000	0.53	0.61	0.71	0.88	0.85	0.57	0.75	0.70	0.94	0.90	0.75	0.68	0.63	0.58	0.62	0.55
10.000	0.39	0.27	0.45	0.32	0.02	0.01	0.02	0.01	1.00	1.01	0.80	0.76	0.56	0.55	0.55	0.52

Minor remarks:

1. Supp. Table 5: the numbers of compounds significantly upregulating miR-132 slightly differ from those mentioned in the text (13 & 11 in the table, 12 & 10 in the text).

Forskolin was considered as a positive control, and thus was not counted. We have rephased the text to avoid any confusion as following: “Besides forskolin, 12 and 10 compounds significantly upregulated miR-132 in primary rat neurons after 24h and 72h, respectively” (Lines 178-179).

2. The culture system employed is not always clearly mentioned. This should be indicated in both the text and figure legends.

The cell culture system employed is now clearly stated in the text (headlines) and figure legends (titles).

3. What do data points represent in Fig. S4A-C? If (some of) these are technical replicates, then they should be averaged instead of being plotted separately.

The data in Fig. S4A-C came from 2 separate experiments, with technical replicates for each experiment.

4. The authors should discuss the fact that Tau phosphorylation did not change upon glycoside treatment in rodent neurons in the context of previous (also from their own work) literature supporting a role for miR-132 in Tau (hyper)phosphorylation.

We indeed did not see a change in pTau: global Tau (Tau5) ratio in rodent neurons. However, as both pTau and Tau5 expressions were reduced, and we generally observed that pTau antibodies gave stronger signals than Tau5 antibodies, we were unsure if this lack of change was real or due to differences in antibody efficiency. There, while we included the result, we found that it would be premature to comment extensively on this observation.

5. Line 284: Reference to Fig. 8 should be reference to Fig. 6.

We have corrected this error.

6. Lines 285-286: digoxin and oleandrin are wrongly indicated as included in Fig. 5D.

We have corrected this error.

7. What is the concentration employed in Fig. 7C?

We included samples of all three dosages to be compared with DMSO. Even though the statistical power was weakened by the intrinsic variance of the dosage gradient, thousands of genes were still significantly changed. We have clarified this design in the figure legend.

8. What was the negative control used for the siRNA transfections?

ON-TARGETplus Non-targeting Control Pool 1 (Horizon Discovery). Information and catalog number of siRNAs and miRNA inhibitors are now included in Table S11.

9. Is there a chance that part of the data provided in Fig. S7 are also shown in Figs. 5 and S6?

The data was obtained on the same set of samples, if that was what the Reviewer asked. The quantification was from new blots, not the same blots.

REVIEWER #2

This paper describes a synergistic small molecule screening and RNA-seq approach to discover compounds that affect microRNA targets. The paper described known RNA modulatory molecules such as cardiac glycosides as molecules that affect specific microRNAs but esp as is the focus here on miR-132 in tau-diseases. I could also see this approach being used for other compounds.

Overall a good paper where strength is not the compound (is is a messy molecule and there is no new target or any target at all) but the biology and perhaps others could use this approach but the approach by itself is not very different than what has been done in other settings.

We thank Reviewer 2 for the suggestion to more accurately describe our screen in the context of existing miRNA screens (now cited and elaborated on) and other high-throughput screens. We also added new data for istaroxime, a non-cardiac glycoside Na⁺/K⁺ inhibitor and referenced studies that cardiac glycosides' IC₅₀ and miR-132 EC₅₀ are similar, again supporting that cardiac glycosides act through their conventional target.

1. Targeting RNA herein is a phenotypic read out and therefore it should be explicitly stated. There is a large interest in RNA targeted drug discovery but it can be in two buckets - one is phenotypic screens and the other is targeting the actual RNA and this paper is clearly a phenotypic screen. as an aside, target centric and systematics screens for chemical mater that target RNA precursors has been done before to inhibit them Velagapudi et al, 2014

We thank the reviewer for this important point that this screen is primarily an unbiased phenotypic screen where the phenotypes are the direct expression levels of miRNAs. The pioneering work of Velagapudi et al, 2014 has been cited in the introduction of the revised manuscript with reference to systematic screens. We modified the introduction to explain our screen in the context of existing screens: “Whereas previous screens focused on a specific miRNA (10, 11, 13, 15) or favored a particular mechanism of action such as direct binding (12, 15), our phenotypic screen is relevant to all miRNAs and inclusive to all mechanisms of action, including direct binding, transcriptional and post-transcriptional modulations, and indirect

regulations. Furthermore, instead of utilizing reporter assays (10, 11, 13), we employed miRNA-sequencing that enabled direct expression profiling of 338 miRNAs for 1370 small molecule compounds.” (Lines 63-69).

2. There is no attempt at mechanism here. Is there any effect on the precursor levels of the microRNA132, is the transcription factor or the DNA's ability to be transcribed affected? These simple studies should be completed. Also, what about the levels of Dicer or Drosha, these are known compounds and some studies on these would be useful. and, explicitly statements about this should be made as it is hard to advance compounds into man without a target and mechanistic rationale. Phenotypic screens for miR effectors have been done before and this should be described see (Young et al, JACS 2010) and this should be described. Also work of Dieters group on miR-21 phenotypic screens in Angewandte should be described also.

For miR-132/212 upregulation, we determined that the mechanism is transcriptional and involves CREB-mediated transcription of the locus. Supporting evidence included: (1) the earlier upregulation of miRNA precursors, (2) the upregulation was blocked by a transcriptional inhibitor, (3) knocking down ATP1A1 or A3 also upregulated miR-132 and -212 and their precursors, (4) cardiac glycosides induced expression of CREB which is a known transcription factor regulating miR-132/212 locus (11, 17). This was not due to a global change in miRNA biogenesis, e.g., via Dicer or Drosha activity, as the levels of other highly expressed miRNAs were not affected either by cardiac glycosides or by knocking down ATP1A1 or ATP1A3 (Fig. S5A, 5F). Both pioneering works from the Dieters group have been described and cited in the Introduction.

3. It is known that miR-132 is affected in heart failure and is affected by an antagomir, see <https://academic.oup.com/eurheartj/article/42/2/178/5974817>. Have the authors considered phenotype readout via gain and loss of function studies with miR-132 to track the outcomes here. If done in a previous studies, please state here

Several prior studies investigated miR-132 in the brain using both gain- and loss-of function approaches. For example, miR-132/212 knockout mice showed reduced basal synaptic transmission (18), deficient spatial learning memory acquisition (19, 20) and impaired tau metabolism (19), but were otherwise fertile and mostly phenotypically normal (18). Treating AD mouse models with miR-132 mimics (21), or viral overexpression of miR-132 (22) rescued behavioral deficits. We are aware of studies implicating a role of miR-132 overexpression in heart diseases, including the study suggested by the Reviewer (23, 24). Excessive miR-132 upregulation in the brain is also associated with seizure (~20-fold) (25). Interestingly, miR-132 is commonly downregulated by ~2-fold in neurodegenerative diseases (Review (26)). Similarly, a 1.5-fold increase in miR-132 expression was associated with improved spatial memory, whereas upregulation of >3-fold inhibited learning (15). Our observation is that cardiac glycosides upregulate miR-132 in neurons by 2~3-fold maximum (Table S6), suggesting that this upregulation is within the physiological range and should be safe. This information has been included in the discussion section in Lines 429-431.

4. what is the conservation of the target through model organisms?

We mentioned in the introduction that miR-132 is fully conserved across mammals, and many of its targets are also highly conserved. Conservation analysis Figure S3 is added to justify our validation of results from human neurons in rodent neurons: “As miR-132 is fully conserved, and many of its targets are highly conserved across mammals (Fig. S3), we also utilized rat cortical neurons for validation.” (Lines 175-176).

5. Are there compounds known that target the ATPase that are not cardiac glycosides that can be tested for these effects? I understand the siRNA was done but chemical inhibition and ablation are two different modes of action. Is the protein target responsible for this activity via non-ATPase roles or not? Also, what is the SAR of this effect on the miR versus the SAR for the canonical target. This should be more clearly stated as a weaker ATP-ase targeting compound should have a weaker effect on the miR if these targets overlap.

We thank the reviewer for the suggestion. We identified a non-cardiac glycoside, istaroxime, that is also an inhibitor of Na⁺/K⁺ ATPases (27). We observed that istaroxime also dose-dependently increased miR-132 and -212 in rat primary neurons, though with uM EC₅₀ instead of sub uM EC₅₀ of CGs (Fig 3B, S4B). The data for istaroxime is plotted specifically here and is mentioned as “[...] istaroxime - a non-cardiac glycoside that also inhibits Na⁺/K⁺ pumps” (Lines 185-186).

We were unable to find a study that measured the IC₅₀ of all the cardiac glycosides tested. The heterogeneity of systems used across studies (e.g., different ATPase subunits – α1, α2, α3; Na⁺/K⁺ ATPases isolated from different species – mouse, human, pig, lamb, dog, shark; different conditions, different cell lines, different methods) prevented a fair comparison and combination of IC₅₀ values from different studies. Nevertheless, generally, EC₅₀ of cardiac glycosides in miR-132 upregulation correlates with their IC₅₀

as ATPases. We listed the IC50 from two studies as well as our miR-132 EC50 below, sorted from smallest to largest below. Proscillaridin A was the most potent in all 3 studies. Therefore, we are confident that the upregulation of miR-132 is primarily through the action of cardiac glycosides on their canonical target. We added to the discussion that “miR-132 EC50 values (Table S6) generally correlate with their reported IC50 values (68, 69), suggesting that their potency can be further improved with structure-activity relationship enhancement.” (Lines 427-429).

Na/K-ATPase from outer medulla of lamb kidneys. Measurement was from ATPase inhibition assay (28)

	IC50 (nM)
Proscillaridin A	0.6
Bufalin	0.77
Cinobufagin	0.78
Digitoxin	3.3
Oleandrin	4.63
Ouabain	5.4
Digoxin	11.5

Average results from 10 human cancer cell lines. Measurement was from decrease in cell viability (29).

	IC50 (nM)
Proscillaridin A	23
Digitoxin	37
Ouabain	78
Digoxin	80

miR-132 upregulation in rat primary neurons. Measurement was from RT-qPCR (this study).

	EC50 (nM)
Proscillaridin A	3.2
Bufalin	46.54
Oleandrin	50.5
Digitoxin	59.83
Digoxin	97.22
Ouabain	118.7
Cinobufagin	196.6

REVIEWER #3

Nguyen et al. conducted a high-throughput chemical screen to identify responsive miRNAs in neurons. Out of several disease-associated miRNAs, the authors focused on miR-132, which was known to be downregulated in ADRD patients. The authors confirmed that several chemicals, including cardiac glycosides, could upregulate pri-miR-132/212 in rodent and human neurons and provide neuroprotective effects. Although the high-throughput small RNA sequencing (RealSeq) coupled with chemical libraries is an exciting and promising strategy, I doubt the impact of this manuscript because of the lack of impactful novel knowledge and translational value. Therefore, I do not recommend the current form of the manuscript for publishing in Nature Communications

While we respectfully disagree with Reviewer 3 that our manuscript “lack of impactful novel knowledge and translational value”, we fully agree that the analyses of the dataset need improvement. For the revision, we perform adjustments that minimize the batch effects and perform additional analyses, including clustering similar compounds and similar miRNAs, leading to a completely revised Figure 2. We are confident that these changes further improve the utility and translational significance of our work to researchers in the field of miRNA therapeutics.

1. One of the possible impacts of this study is to provide a resource to study interactions between miRNAs and small molecules in neurons. However, in the current form of the manuscript, the authors did not thoroughly analyze the whole dataset. For example, a heatmap comparing all miRNAs and all compounds (or compound families) was missing in Fig 2. Therefore, Fig 2A-F looks like cherry-picking examples. The authors should provide an unbiased interpretation of the high-

throughput data in Fig 2 before moving on to the miR-132 part. Just putting RNA-seq read counts in Supplementary Table is not sufficient.

We agree with the Reviewer's viewpoint that conducting more comprehensive analyses of our large dataset will significantly enhance the impact of this study while also serving as a valuable resource for the community. To address this suggestion, we have performed more extensive analyses, including minimizing the batch effect and clustering analyses for 300+ miRNAs expressed in human neurons. Based on the observed effects on miRNome, the compounds were categorized in an unsupervised way into five clusters, and miRNAs were grouped into four clusters. The compounds with shared physical/chemical properties or those influencing common functional pathways exhibited similar impacts on miRNA sets. The main results of this work are now integrated into new Figures 2 and S2 and described in Lines 125-169 of the revised manuscript.

2. Because of the batch effect (Fig S2), the authors normalized RPM with DMSO controls in the same batch. However, the authors used the miR-132 plate rank, rather than normalized RPM, to analyze the dataset (Line 161). I wonder whether the normalization method did not work to compare different plates. If the normalization did not work well, how can we interpret Figures 2E and F? Thus, several control plots are needed to support whether the normalization significantly compromised the batch effect. For example, how are the expression level of housekeeping miRNAs (Fig 1D) before and after normalization?

In the revised version of manuscript, we have corrected the batch effect using the ComBat algorithm (Figure S2). Below shown the expression levels of housekeeping miRNAs before and after ComBat correction. This strategy allowed us to perform more accurate analyses across batches and provide additional insights. The corrected value of each miRNA in each sample is also provided in the new Supplementary Table 3.

3. Another major issue is the lack of impressive new knowledge.

We respectfully disagree with the Reviewer for two reasons. First, our study is the first to discover and validate small molecule drugs that upregulate miR-132, one of the most consistently downregulated miRNAs in Alzheimer's disease and other neurodegenerative diseases that is also recognized as a neuroprotective miRNA (26, 30, 31). We also elucidated the mechanism of miR-132 upregulation induced by CG, attributing it to the inhibition of Na⁺/K⁺ ATPases. These new findings support the potential optimization of cardiac glycosides for neurodegenerative diseases. Worth noting, neuron-

enriched Na⁺/K⁺ α3 isoform has been implicated in many neurodevelopmental disorders(32), but little is known about its role in neurodegenerative diseases. Our study suggests that ATP1A1 and ATP1A3 dysregulation, contributing to miR-132 reduction, may play a role in neurodegenerative diseases. We added to the discussion that: “Second, while dysfunctions in ubiquitous ATP1A1 and neuron-specific ATP1A3 have been predominantly linked to neurodevelopmental disorders (74-76), their role in regulating miR-132 may suggest potential underexplored functions in neurodegenerative diseases.” (Lines 448-450).

Second, we would like to argue that the screen itself provides substantial new knowledge for researchers striving to identify small molecule compounds regulating specific miRNAs. This is the first high-throughput screen of 1370 small molecules encompassing the entire miRNome as a readout. Prior efforts focused on identification of small molecule regulators of a single miRNA, and only a few such small molecules have been reported thus far. Our approach establishes a searchable database for diverse follow-up studies on hundreds of neuronal miRNAs, and a proof-of-principle for this screening strategy to be explored more broadly. While we highlight the limitations of the screen (Lines 399-411) and focus this study on miR-132, the key miRNA associated with neurodegenerative diseases, we have since validated some compounds upregulating another miRNA, miR-124, that promotes neuronal differentiation (33, 34). To illustrate, we successfully validated the impact of the compound PP121 in boosting miR-124 levels in neuroblastoma SKNAS cells (as depicted in the figure below), rat primary neurons, and other cell lines (data not shown).

Treatment of SKNAS neuroblastoma cells with PP121, and then bufalin to also upregulate miR-132, converted SKNAS cells from a fibroblast-like morphology to a neuronal-like morphology, therefore supporting a novel chemical protocol of neuro-differentiation (to be reported in a separate manuscript). We use this as an example showing how our dataset can be utilized to discover small molecule compounds that regulate miRNAs of biological and/or clinical promise. Alternatively, the dataset can also be used to discover what miRNAs maybe affected by a set of compounds acting on a specific target or pathway, e.g. Fig. 2D-G. With the validation rate of 0.14%, which is similar to or exceeding standard phenotypic screens (1-3), we are convinced that our screen provides a valuable resource for hypotheses generation and multiple follow-up studies. As such, this dataset can indeed inform new discoveries.

4. The manuscript starts with a screening of miR-132-modulating chemicals but ends with an ambiguous contribution of miR-132 in the neuroprotective effect of cardiac glycosides. As the authors mentioned, (“...Further investigation is needed to determine the contribution of miR-132 upregulation to Tau downregulation and neuroprotection...”; Line 375), the current form of the manuscript does not clearly interpret the contribution of miR-132 out of the enormous

transcriptomic effects (Fig 7) of cardiac glycosides. However, the neuroprotective effect of cardiac glycosides in ADRD and tauopathies was already shown in recent papers (Ref. 56 and 57). The authors may investigate the contribution of miR-132 in the context of cardiac glycosides by including a miR-132 LNA inhibitor in Fig 4L-O. Also, the contribution of miR-132 can be analyzed by grouping differentially expressed genes with and without the miR-132 seed sequence in Fig 7C. However, I think the contribution of miR-132 would be minor based on the huge transcriptomic changes in Fig 7C. If miR-132 has a critical neuroprotective effect, forskolin (Fig 3A) should show a similar result in Fig 4-6.

We agree with the Reviewer that the effects of cardiac glycosides are unlikely to be mediated entirely via miR-132/212 upregulation. To better address this point, we performed the rescue experiment and showed that the miR-132/212 inhibitors partially reduced the neuroprotective effect of cardiac glycosides, indicating that it is at least partially mediated by miR-132/212 (see pages 8 of this response letter, and Fig. 4O of the revised manuscript). Of note, these experiments on highly sensitive neuronal cultures combine three insults (Ab, CGc, and miR-132 oligonucleotide inhibitors), each with a restricted window of activity and some level of toxicity, making them quite challenging. Nevertheless, the results support our conclusions.

In addition, the key miR-132 targets implicated in neuroprotection, tau, and Ab homeostasis (e.g., FOXO3a, RBFOX1, TMEM106B, ITBPKB) and other high-confidence miR-132 targets are downregulated by cardiac glycosides. Since, indeed, the CGs induce massive transcriptomic changes, only partly mediated by miR-132, we wrote that "Our transcriptomic results support that many neuronal pathways are altered, suggesting that cardiac glycosides can modulate multiple pathways that converge on the downregulation of Tau and increase neuroprotection." (Lines 453-456).

We refrained from further experiments with forskolin because its effects on miR-132 have been previously established (10, 11, 17) and it is also known to broadly affect many additional pathways (12) (also see response to Reviewer 1, page 5).

Minor comments

1. Fig 1E. Show other control (housekeeping) miRNAs on the same plot.

We have replotted the Fig. 1E following the Reviewer's suggestion.

2. Line 143. I don't think 'COX inhibitors GENERALLY downregulated miR-26b'. Only SOME COX inhibitors downregulated miR-26b.

We replotted Fig. 2 with additional data analyses, and revised the text for accuracy. COX inhibitors were no longer shown.

3. Fig 2F, H and Line 153. In many cases, miRNAs with similar sequences have similar read counts because of sequencing errors and mapping problems. I'm not sure whether the reads of miR-376c-3p, miR-376a-3p, and miR-376-b-3p are uniquely counted during analysis. Also, the same issue is raised to miR-103a-3p, miR-103b, and the let-7 family.

We acknowledge that distinguishing miRNAs with similar sequences poses a challenge in achieving high-confidence discrimination through RNAseq. This issue is well-recognized in the field, and currently, no flawless solution exists. To minimize this, we employ specific alignment parameters, based on Ziemann et al., 2016 (35). The parameters used in this study include bowtie -S --chunkmbs 512 -p 4 -n 1 -l 17 -q -m 25 -k 1 --best --strata. In our dataset, we observed major differences in the expression of these closely

related miRNAs and their levels exhibit a lack of correlation (Fig. 2D, Supplementary Table 2). More importantly, we refrain from making specific conclusions pertaining to individual miRNAs within the same family.

4. Fig 2G-H. Is this data from DMSO-treated samples? I couldn't find a sufficient description of the analysis method for these Figures.

The corresponding panel (Fig. 2L in revised version) was based on all sequenced samples. We have clarified this information in the revised figure legend.

5. miR-132-3p and miR-212-3p share the seed sequence (<https://mirgenedb.org/show/hsa/Mir-132-P1>; <https://mirgenedb.org/show/hsa/Mir-132-P2>). Thus, not only miR-132 but also miR-212 may contribute to the target repression.

We agree that miR-132 and miR-212 might function synergically as they share the seed sequence and are co-expressed from the same locus. Nevertheless, miR-132 is believed to be the major player since its abundance in neurons is much higher than miR-212. We discussed miR-212 as: “miR-212, which shares the seed sequence with miR-132 and is co-expressed from the same locus” (Lines 191-192) and “While we focus on miR-132 due to its great abundance (Tables S1, (39)), many of the effects observed may also be facilitated by miR-212 upregulation.” (Lines 413-415)

6. "pre_miR132_R" binds to pri-miR-132 but not to pre-miR-132. Also, "pre_miR212_R" binds to only pri-miR-212. Throughout the manuscript, change pre-miR-132 and pre-miR-212 into pri-miR-132 and pri-miR-212. Also, do not use 'precursor' but 'primary transcript.'

Our original description is accurate as we used PCR primers amplifying pre-miR-132 stem-loop sequence designed in a prior study (36). Similarly, we utilized primers amplifying pre-miR-212. We verified that the primer pairs amplified both human and rodent pre-miR132 and pre-miR-212. [The sequence for human pre-miR-132 is shown below. The mature transcripts miR-132-3p and miR-132-5p are capitalized in blue. The binding sites for the primers are highlighted in yellow.] Of note, both *pre*-miR-132 and *pri*-miR-132 could be amplified in these reactions. However, additional reactions with several alternative pairs of primers designed to amplify primary pri-miR-132/212 transcripts more specifically exhibited CT values >35, suggesting very low levels, likely due to the efficient processing by the endoribonuclease Drosha (37). Please note that, nevertheless, our data (Fig. 3D-E, S5B-D) indicate that cardiac glycosides regulate miR-132 transcription rather than processing. We add the clarification that “In addition, although cardiac glycosides activated transcription in the miR-132/212 locus, the primary pri-miR-132/212 transcripts were barely detectable, likely due to the efficient processing by the endoribonuclease Drosha (73).” (Lines 445-447)

Description: *Homo sapiens* hsa-mir-132 precursor miRNA

ccgccc**ccgcgucuccagggca****ACCGUGGCCUUUCGAUUGUUACU**gugggaacuggagg**UAACAGUCUAC**
AGCCAUGGUCGccccgcagcaccccacgcgc

7. Line 192. Briefly explain why the authors used a CREB inhibitor here.

We briefly explained the use of a CREB inhibitor: “CREB is a known transcriptional activator of the miR-132/212 locus (27-29), and we hypothesized that the identified compounds regulate miR-132/212 via CREB. Indeed, the upregulation of miR-132 by various compounds was completely blocked by

pretreatment with the transcription inhibitor actinomycin D and a CREB inhibitor (Figs. 3D-E and S5B-D).” (Lines 204-208).

8. Fig 5. The authors utilized only one cell line for WT and another for P301L neurons. I wonder whether the cell viability difference between WT and P301L neurons upon the treatment of cardiac glycosides can be reproduced in other iPSC neurons.

A difference in cardiac glycosides toxicity between WT and Tau-P301L neurons (Fig. 6a-f), was seen only after 72h treatment and corresponded to a maximum 20-30% loss in viability. The fact that this was seen at 72h but not 24h might have to do, not with the direct targets of the small molecules, but consequences of secondary and off target mechanisms. On one hand, this can be due to differences in genomic background or even artifacts between independent iPSC and derived neuronal cell lines. On the other hand, patient-derived neurons at this stage of differentiation already have mutant-tau dependent changes to the proteostasis network (“sensitized neurons”), such as activation of stress responses (8, 9), that may influence the response to the small molecules. To answer the Reviewer’s comment, multiple cell lines carrying the same and different mutations, as well as WT controls and isogenic WT controls would be needed, which, however, is beyond the scope of this manuscript. We added that “**Whether the difference in sensitivity to cardiac glycosides between WT and Tau-P301L neurons can be reproduced in other human iPSC-neuron lines bearing disease-relevant Tau mutations remain to be tested.**” (Lines 309-311) to caution against premature generalization of our data.

Typo and simple adjustment

1. Fig 2A-D. Include the number of genes per category on the Y axis in the legend.

These panels have been removed in the revised Fig. 2.

2. Fig 2E-F. The X-axis labeling does not match the number of data points.

The labeling in the corresponding panel (Fig. 2D-G) has been clarified.

3. Line 71. Ref.22 seems like a wrong reference in this context.

We have corrected this error.

4. Fig 1C, E. Indicate the number of miRNAs and drugs on the X axis.

We have made the corresponding modifications in Fig. 1.

5. Fig 2E-H. Label the color bar.

Color bar has been added in Fig. 2.

6. Line 152. three transcripts  three miRNAs

The related sentence has been rephrased in the revised manuscript.

7. Fig 3B, Fig S3A. It is difficult to interpret because of too much data in a single plot.

We have enlarged the corresponding panel to allow clear visualization.

8. Line 225. Fig 4I -> 4L

We have corrected this error.

References cited in the response to reviewers

1. H. Majd *et al.*, Deriving Schwann cells from hPSCs enables disease modeling and drug discovery for diabetic peripheral neuropathy. *Cell Stem Cell* **30**, 632-647 e610 (2023).
2. O. E. Reyes Gaido *et al.*, An improved reporter identifies ruxolitinib as a potent and cardioprotective CaMKII inhibitor. *Sci Transl Med* **15**, eabq7839 (2023).
3. X. Liu *et al.*, ARIH1 activates STING-mediated T-cell activation and sensitizes tumors to immune checkpoint blockade. *Nat Commun* **14**, 4066 (2023).
4. M. J. Van Kanegan *et al.*, BDNF mediates neuroprotection against oxygen-glucose deprivation by the cardiac glycoside oleandrin. *J Neurosci* **34**, 963-968 (2014).
5. D. E. Dunn *et al.*, In vitro and in vivo neuroprotective activity of the cardiac glycoside oleandrin from Nerium oleander in brain slice-based stroke models. *J Neurochem* **119**, 805-814 (2011).
6. S. Eid *et al.*, Identification of a Cardiac Glycoside Exhibiting Favorable Brain Bioavailability and Potency for Reducing Levels of the Cellular Prion Protein. *Int J Mol Sci* **23**, (2022).
7. D. S. Hong *et al.*, First-in-human study of pbi-05204, an oleander-derived inhibitor of akt, fgf-2, nf-kappaBeta and p70s6k, in patients with advanced solid tumors. *Invest New Drugs* **32**, 1204-1212 (2014).
8. M. C. Silva *et al.*, Human iPSC-Derived Neuronal Model of Tau-A152T Frontotemporal Dementia Reveals Tau-Mediated Mechanisms of Neuronal Vulnerability. *Stem Cell Reports* **7**, 325-340 (2016).
9. M. C. Silva *et al.*, Prolonged tau clearance and stress vulnerability rescue by pharmacological activation of autophagy in tauopathy neurons. *Nat Commun* **11**, 3258 (2020).
10. M. E. Klein *et al.*, Homeostatic regulation of MeCP2 expression by a CREB-induced microRNA. *Nat Neurosci* **10**, 1513-1514 (2007).
11. N. Vo *et al.*, A cAMP-response element binding protein-induced microRNA regulates neuronal morphogenesis. *Proc Natl Acad Sci U S A* **102**, 16426-16431 (2005).
12. T. Pullaiah, in *Forskolin: Natural Sources, Pharmacology and Biotechnology*, T. Pullaiah, Ed. (Springer Nature Singapore, Singapore, 2022), pp. 65-106.
13. C. Sato *et al.*, Tau Kinetics in Neurons and the Human Central Nervous System. *Neuron* **97**, 1284-1298 e1287 (2018).
14. E. Patrick *et al.*, Dissecting the role of non-coding RNAs in the accumulation of amyloid and tau neuropathologies in Alzheimer's disease. *Mol Neurodegener* **12**, 51 (2017).
15. K. F. Hansen *et al.*, miRNA-132: a dynamic regulator of cognitive capacity. *Brain Struct Funct* **218**, 817-831 (2013).
16. E. Salta, A. Sierksma, E. Vanden Eynden, B. De Strooper, miR-132 loss de-represses ITPKB and aggravates amyloid and TAU pathology in Alzheimer's brain. *EMBO Mol Med* **8**, 1005-1018 (2016).
17. A. J. van Zonneveld *et al.*, MicroRNA-132 regulates salt-dependent steady-state renin levels in mice. *Commun Biol* **3**, 238 (2020).
18. J. Remenyi *et al.*, miR-132/212 knockout mice reveal roles for these miRNAs in regulating cortical synaptic transmission and plasticity. *PLoS One* **8**, e62509 (2013).
19. J. Hernandez-Rapp *et al.*, Memory formation and retention are affected in adult miR-132/212 knockout mice. *Behav Brain Res* **287**, 15-26 (2015).
20. K. F. Hansen *et al.*, Targeted deletion of miR-132/-212 impairs memory and alters the hippocampal transcriptome. *Learn Mem* **23**, 61-71 (2016).
21. P. Y. Smith *et al.*, miR-132/212 deficiency impairs tau metabolism and promotes pathological aggregation in vivo. *Hum Mol Genet* **24**, 6721-6735 (2015).
22. H. Walgrave *et al.*, Restoring miR-132 expression rescues adult hippocampal neurogenesis and memory deficits in Alzheimer's disease. *Cell Stem Cell* **28**, 1805-1821 e1808 (2021).
23. A. Foinquinos *et al.*, Preclinical development of a miR-132 inhibitor for heart failure treatment. *Nat Commun* **11**, 633 (2020).

24. J. Taubel *et al.*, Novel antisense therapy targeting microRNA-132 in patients with heart failure: results of a first-in-human Phase 1b randomized, double-blind, placebo-controlled study. *Eur Heart J* **42**, 178-188 (2021).
25. E. M. Jimenez-Mateos *et al.*, miRNA Expression profile after status epilepticus and hippocampal neuroprotection by targeting miR-132. *Am J Pathol* **179**, 2519-2532 (2011).
26. E. Salta, B. De Strooper, microRNA-132: a key noncoding RNA operating in the cellular phase of Alzheimer's disease. *FASEB J* **31**, 424-433 (2017).
27. S. Aditya, A. Rattan, Istaroxime: A rising star in acute heart failure. *J Pharmacol Pharmacother* **3**, 353-355 (2012).
28. S. Paula, M. R. Tabet, W. J. Ball, Jr., Interactions between cardiac glycosides and sodium/potassium-ATPase: three-dimensional structure-activity relationship models for ligand binding to the E2-Pi form of the enzyme versus activity inhibition. *Biochemistry* **44**, 498-510 (2005).
29. S. Johansson *et al.*, Cytotoxicity of digitoxin and related cardiac glycosides in human tumor cells. *Anticancer Drugs* **12**, 475-483 (2001).
30. P. Lau *et al.*, Alteration of the microRNA network during the progression of Alzheimer's disease. *EMBO Mol Med* **5**, 1613-1634 (2013).
31. R. El Fatimy *et al.*, MicroRNA-132 provides neuroprotection for tauopathies via multiple signaling pathways. *Acta Neuropathol* **136**, 537-555 (2018).
32. A. Vezyroglou *et al.*, The Phenotypic Continuum of ATP1A3-Related Disorders. *Neurology* **99**, e1511-e1526 (2022).
33. A. S. Yoo *et al.*, MicroRNA-mediated conversion of human fibroblasts to neurons. *Nature* **476**, 228-231 (2011).
34. E. V. Makeyev, J. Zhang, M. A. Carrasco, T. Maniatis, The MicroRNA miR-124 promotes neuronal differentiation by triggering brain-specific alternative pre-mRNA splicing. *Mol Cell* **27**, 435-448 (2007).
35. M. Ziemann, A. Kaspi, A. El-Osta, Evaluation of microRNA alignment techniques. *RNA* **22**, 1120-1138 (2016).
36. G. A. Wayman *et al.*, An activity-regulated microRNA controls dendritic plasticity by down-regulating p250GAP. *Proc Natl Acad Sci U S A* **105**, 9093-9098 (2008).
37. T. C. Chang, M. Pertea, S. Lee, S. L. Salzberg, J. T. Mendell, Genome-wide annotation of microRNA primary transcript structures reveals novel regulatory mechanisms. *Genome Res* **25**, 1401-1409 (2015).

REVIEWER COMMENTS

Reviewer #1 (Remarks to the Author):

I appreciate the efforts of the authors to address all issues and points of discussion in their rebuttal and revised manuscript by adding new data and discussion. However, there are some remaining issues that need to be clarified.

1. It has to be clearly stated that ouabain is not clinically approved or utilized in recent clinical trials.
2. With respect to my previous remark (#5 in rebuttal) on the therapeutic/toxicity window, most of the compounds induce a significant reduction of neuronal viability (Fig. 6A-F) and concentrations indicated as part of the yellow boxes in Fig. 5 (down to 10-3 uM). I understand that the authors define 'a <30% loss of at least two synaptic/microtubule markers' (Fig. 5) and a viability reduction of ~20% (Fig. 6) as acceptable, however, I believe that more evidence is required (from own data or the literature) to support the putative clinical relevance of these margins. This is also related to remark #7 in the rebuttal: what is the evidence supporting the assumption that a significant loss of viability of <15% is of no biological (and subsequently clinical) importance. The authors should at least discuss this further in the text.
3. The authors have misunderstood my previous remark (#16 in rebuttal) about the quantification of total Tau levels based on western blot data. I do not object to the usage of the term 'total Tau'. Instead, I did not think that all the (non-phosphorylated) isoforms of 'total Tau' were effectively taken into consideration, as the cropped Tau blots (including those of the Tau5 Ab) only depict two bands. However, I acknowledge the authors' justification that these cells do not express multiple Tau isoforms. There is no need to change the term 'total Tau' to 'global Tau'. Nevertheless, providing uncropped blots for these results would be sufficient to clarify this point.
4. Re-analyzing samples starting from the same RNA (or protein) input cannot be considered as a valid experimental replicate. These data should be removed from any quantifications.
5. The numbers of predicted miR-132 targets separately among up- and down- regulated DEGs should be provided. In addition, the authors should provide some discussion on the low proportion of miR-132 predicted targets among DEGs.
6. Plotting individual technical replicates as separate data points is not a proper mode of performing statistics. Technical replicates should collapse onto one single data point per individual experimental replicate (e.g. Fig. S6A-C). Statistics (and conclusions) should be adapted accordingly.

Reviewer #2 (Remarks to the Author):

I do not have any additional comments, the authors have done an exceptional job of addressing all comments. As far as I am concerned this should be accepted now

Reviewer #3 (Remarks to the Author):

I am happy with the quality of the revised manuscript, especially Fig 2 (comprehensive analysis) and Fig 4O (rescue experiment). Also, the example of miR-124 proved the value of the HTS data as a resource to readers. In addition, the control plot in Fig 1E also showed the clear up-regulation of miR-132 in the dataset. Therefore, I recommend the manuscript be published in Nature Communications.

However, the authors still need to correct the pre-miRNA issue. I think the authors were confused pre-miRNA with the miRBase-annotated 'stem-loop' (<https://mirbase.org/hairpin/MI0000449>). The 'stem-loop' is an arbitrarily defined partial pri-miRNA sequence. More specifically, the 5' end of pre-miRNA should be the same as the 5' end of 5p mature miRNA. Also, the 3' end of pre-miRNA should be the same as the 3' end of 3p mature miRNA. In this case, the 5' end of pre-miR-132 is 'ACCGUGG' rather than 'ccgccc'. Please note that DICER cuts only the terminal loop (for example, for miR-132: 'gugggaacuggagg'). The forward PCR primer cannot bind to the pre-miRNA because only two nucleotide overlaps exist. The authors must amend all figures and text, including the Discussion part.

Please note that significant text changes are colored blue in both the response and the revised manuscript.

Reviewer #1 (Remarks to the Author):

I appreciate the efforts of the authors to address all issues and points of discussion in their rebuttal and revised manuscript by adding new data and discussion. However, there are some remaining issues that need to be clarified.

1. It has to be clearly stated that ouabain is not clinically approved or utilized in recent clinical trials.

We added in the result section that “Of note, digoxin is clinically approved for treating various heart conditions, whereas ouabain is not clinically approved or utilized in recent clinical trials.” (Lines 182-184).

2. With respect to my previous remark (#5 in rebuttal) on the therapeutic/toxicity window, most of the compounds induce a significant reduction of neuronal viability (Fig. 6A-F) and concentrations indicated as part of the yellow boxes in Fig. 5 (down to 10-3 uM). I understand that the authors define ‘a <30% loss of at least two synaptic/microtubule markers’ (Fig. 5) and a viability reduction of ~20% (Fig. 6) as acceptable, however, I believe that more evidence is required (from own data or the literature) to support the putative clinical relevance of these margins. This is also related to remark #7 in the rebuttal: what is the evidence supporting the assumption that a significant loss of viability of <15% is of no biological (and subsequently clinical) importance? The authors should at least discuss this further in the text.

From experience, for this specific viability assay and variability of measurements on human iPSC-derived neurons within technical replicates, including for DMSO-treated wells, a 10-15% variability is often obtained, and it doesn’t signify variability on neuronal health/death, but rather differences across neuronal wells cultured for many weeks, subjected to media changes and washes. This introduces technical variation, and based on this experience, we look at changes in viability larger than 10-15% as being more significantly caused by a compound or agent, above technical variability. The yellow boxes do not signify “acceptable toxicity” but rather experimental conditions with a stated, measured low or very low toxicity. There is no available extrapolation between iPSC neuronal toxicity and clinical safety. A low toxicity profile is only considered based on %loss vs. the existing variability inherent to culture conditions and the assay itself. To clarify this point to readers, we added that: “This reduction is in close proximity to the technical variability typically observed in long-term cultures of iPSC-neurons.” (Lines 330-331).

3. The authors have misunderstood my previous remark (#16 in rebuttal) about the quantification of total Tau levels based on western blot data. I do not object to the usage of the term ‘total Tau’. Instead, I did not think that all the (non-phosphorylated) isoforms of ‘total Tau’ were effectively taken into consideration, as the cropped Tau blots (including those of the Tau5 Ab) only depict two bands. However, I acknowledge the authors’ justification that these cells do not express multiple Tau isoforms. There is no need to change the term ‘total Tau’ to ‘global Tau’. Nevertheless, providing uncropped blots for these results would be sufficient to clarify this point.

We thank the reviewer for the clarification. The term “global tau” is changed back to “total tau”. Uncropped Western blots for Tau 5 are now included in the source data for Figures 5, S8, and S9. One blot is provided below as an example to show that the two bands shown in the figure are the two dominant bands.

4. Re-analyzing samples starting from the same RNA (or protein) input cannot be considered as a valid experimental replicate. These data should be removed from any quantifications.

We collapsed the technical replicates in Figures 5 and S8. The source data and figure legend were also updated accordingly.

5. The numbers of predicted miR-132 targets separately among up- and down-regulated DEGs should be provided. In addition, the authors should provide some discussion on the low proportion of miR-132 predicted targets among DEGs.

The number of predicted miR-132 targets that were down or upregulated in our RNA-seq is provided in the table below and the full list is included in the updated Table S7. We added to the discussion that: "The relatively low proportion of miR-132 predicted targets among the downregulated genes could be due to cascades of downstream regulated genes, as well as multiple pathways affected by the compounds." (Lines 353-355).

Database predicted targets	Downregulated			Upregulated		
	Digoxin	Oleandrin	Proscillaridin A	Digoxin	Oleandrin	Proscillaridin A
miRTarBase	12	17	12	30	31	28
TargetScan	24	30	20	35	30	28
miRDB	35	46	33	48	57	38

6. Plotting individual technical replicates as separate data points is not a proper mode of performing statistics. Technical replicates should collapse onto one single data point per individual experimental replicate (e.g. Fig. S6A-C). Statistics (and conclusions) should be adapted accordingly.

As mentioned in response to point 4, we collapsed the technical replicates in Figures 5 and S8. The source data and figure legend were also updated accordingly.

Reviewer #2 (Remarks to the Author):

I do not have any additional comments, the authors have done an exceptional job of addressing all comments. As far as I am concerned this should be accepted now

Reviewer #3 (Remarks to the Author):

I am happy with the quality of the revised manuscript, especially Fig 2 (comprehensive analysis) and Fig 4O (rescue experiment). Also, the example of miR-124 proved the value of the HTS data as a resource to readers. In addition, the control plot in Fig 1E also showed the clear up-regulation of miR-132 in the dataset. Therefore, I recommend the manuscript be published in Nature Communications.

However, the authors still need to correct the pre-miRNA issue. I think the authors were confused pre-miRNA with the miRBase-annotated 'stem-loop' (<https://mirbase.org/hairpin/MI0000449>). The 'stem-loop' is an arbitrarily defined partial pri-miRNA sequence. More specifically, the 5' end of pre-miRNA should be the same as the 5' end of 5p mature miRNA. Also, the 3' end of pre-miRNA should be the same as the 3' end of 3p mature miRNA. In this

case, the 5' end of pre-miR-132 is 'ACCGUGG' rather than 'ccgcccc'. Please note that DICER cuts only the terminal loop (for example, for miR-132: 'gugggaacuggagg'). The forward PCR primer cannot bind to the pre-miRNA because only two nucleotide overlaps exist. The authors must amend all figures and text, including the Discussion part.

We apologize for the mistake on our part and thank the reviewer for noticing and correcting it. All labels of “pre-miR-132”, “pre-miR-212”, and “precursors” were changed to “pri-miR-132”, “pri-miR-212”, and “primary transcript” respectively. The erroneous discussion “In addition, although cardiac glycosides activated transcription in the miR-132/212 locus, the primary pri-miR-132/212 transcripts were barely detectable, likely due to the efficient processing by the endoribonuclease Drosha (73)” has been removed.

REVIEWERS' COMMENTS

Reviewer #1 (Remarks to the Author):

The authors have addressed all my additional remarks.

There is only one pending point: In the sentence "This reduction is in close proximity to the technical variability typically observed in long-term cultures of iPSC-neurons." (Lines 330-331), the authors should indicate in parentheses both the reduction of viability observed in the current study and the expected range of viability reduction due to technical variability.

REVIEWER 1

The authors have addressed all my additional remarks.

There is only one pending point: In the sentence "This reduction is in close proximity to the technical variability typically observed in long-term cultures of iPSC-neurons." (Lines 330-331), the authors should indicate in parentheses both the reduction of viability observed in the current study and the expected range of viability reduction due to technical variability.

We amended the text to describe the technical variability with iPSC models more accurately: "Interestingly, in Tau-P301L neurons, the toxicity observed was minimal, with <15% viability loss at the highest concentrations at 72h (Fig. 6D-F). Given the inherent technical variability across culture wells of iPSC-neurons, cultured for >6 weeks of differentiation, which results in viability reads within 5-10% variability across replicates (44, 47), and accounting for the standard deviation error within the assay (5-10%), these results suggest a negligible effect on P301L neurons viability at 72h treatment. These results were consistent with the previous immunoblot data (Fig. 5G-R), showing that P301L neurons were more resistant to cardiac glycoside toxicity than WT neurons." (Lines 323-330).